# Single-cell multiomics analysis of chronic myeloid leukemia links cellular heterogeneity to therapy response

Rebecca Warfvinge[1], Linda Geironson Ulfsson[1], Parashar Dhapola[1], Fatemeh Safi[1], Mikael Sommarin[1], Shamit Soneji[1], Henrik Hjorth-Hansen[2,3], Satu Mustjoki[4,5,6], Johan Richter[7,8], Ram Krishna Thakur[1]*, Göran Karlsson[1]*

[1]Division of Molecular Hematology, Lund Stem Cell Center, Lund University, Lund, Sweden; [2]Department of Hematology, St Olavs Hospital, Trondheim, Norway; [3]Department of Cancer Research and Molecular Medicine, Norwegian University of Science and Technology (NTNU), Trondheim, Norway; [4]Translational Immunology Research Program and Department of Clinical Chemistry and Hematology, University of Helsinki, Helsinki, Finland; [5]Hematology Research Unit Helsinki, Helsinki University Hospital Comprehensive Cancer Center, Trondheim, Norway; [6]iCAN Digital Precision Cancer Medicine Flagship, Helsinki, Finland; [7]Division of Molecular Medicine and Gene Therapy, Lund Stem Cell Center, Lund University, Lund, Sweden; [8]Department of Hematology, Oncology and Radiation Physics, Skåne University Hospital, Lund, Sweden

*For correspondence:
ram_krishna.thakur@med.lu.se
(RKrishnaT);
goran.karlsson@med.lu.se (GK)

## eLife assessment

This study presents **fundamental** insights into the heterogeneity of chronic myeloid leukemia (CML) stem cells and their response to tyrosine kinase inhibitor therapy, shedding light on potential mechanisms underlying treatment failure. The study's robust methodology, supported by validation with bulk RNA-seq data and surface marker analysis, provides **compelling** evidence for the identified associations between cellular composition and treatment outcome. These findings contribute to our understanding of CML pathogenesis and may inform the development of more targeted therapeutic strategies.

**Abstract** The advent of tyrosine kinase inhibitors (TKIs) as treatment of chronic myeloid leukemia (CML) is a paradigm in molecularly targeted cancer therapy. Nonetheless, TKI-insensitive leukemia stem cells (LSCs) persist in most patients even after years of treatment and are imperative for disease progression as well as recurrence during treatment-free remission (TFR). Here, we have generated high-resolution single-cell multiomics maps from CML patients at diagnosis, retrospectively stratified by BCR::ABL1$^{IS}$ (%) following 12 months of TKI therapy. Simultaneous measurement of global gene expression profiles together with >40 surface markers from the same cells revealed that each patient harbored a unique composition of stem and progenitor cells at diagnosis. The patients with treatment failure after 12 months of therapy had a markedly higher abundance of molecularly defined primitive cells at diagnosis compared to the optimal responders. The multiomic feature landscape enabled visualization of the primitive fraction as a mixture of molecularly distinct BCR::ABL1$^+$ LSCs and BCR::ABL1$^-$ hematopoietic stem cells (HSCs) in variable ratio across patients, and guided their prospective isolation by a combination of CD26 and CD35 cell surface markers. We for the first time show that BCR::ABL1$^+$ LSCs and BCR::ABL1$^-$ HSCs can be distinctly separated as CD26$^+$CD35$^-$ and CD26$^-$CD35$^+$, respectively. In addition, we found the ratio of LSC/HSC to be higher

in patients with prospective treatment failure compared to optimal responders, at diagnosis as well as following 3 months of TKI therapy. Collectively, this data builds a framework for understanding therapy response and adapting treatment by devising strategies to extinguish or suppress TKI-insensitive LSCs.

## Introduction

The persistence, burden, and evolvability of LSCs during therapy provide common threads to address the major challenges remaining in CML (*Cortes et al., 2021*; *Michor et al., 2005*). For example, the acquisition of the BCR::ABL1 kinase domain (KD), and additional somatic mutations within LSCs could either necessitate a switch to another TKI or precipitate a blast crisis (*Bolton-Gillespie et al., 2013*; *Giustacchini et al., 2017*; *Nieborowska-Skorska et al., 2012*; *O'Hare et al., 2007*; *Sorel et al., 2004*). Moreover, even in the absence of KD mutations, the LSC burden at diagnosis is linked to cytogenetic and molecular response in the chronic phase (*Khorashad et al., 2006*; *Mustjoki et al., 2013*), and their presence is associated with recurrence during TFR (*Pagani et al., 2020*), thus compelling life-long treatment in the majority of patients.

The de facto standard for disease monitoring, BCR::ABL1 transcript/gDNA quantification using qPCR, Sanger, and/or Next Generation Sequencing (NGS), however, predictably only captures cells positive for BCR::ABL1 expression and translocation and does not inform the number and identity of fully leukemogenic cells (*Radich et al., 2019*). This is pertinent given that a combination of cell sorting and BCR::ABL1 quantification suggested that the presence of BCR::ABL1 signal in the stem cell compartment rather than the lymphocytes correlates with TFR loss (*Kinstrie et al., 2020*; *Pagani et al., 2020*). Using HSC specific markers, however, is confounded by the observations that in remission, patients in chronic phase restore Philadelphia chromosome-negative (Ph⁻) hematopoiesis (*Bergamaschi et al., 1994*; *Carella et al., 1993*; *Claxton et al., 1992*; *Coulombel et al., 1983*; *Deininger, 2003*), implying the co-existence of BCR::ABL1⁻ HSCs and BCR::ABL1⁺ LSCs. However, because their proportion varies across individuals and treatment stages (*Mustjoki et al., 2013*), and both LSCs and HSCs reside within similar immunophenotypic compartments (Lin⁻CD34⁺CD38⁻/low) (*Eisterer et al., 2005*; *Petzer et al., 1996*), discrimination between these populations – and thereby reliably estimating LSCs – remains difficult. As a result, understanding CML LSC identity, heterogeneity, and vulnerability to TKI therapy remain outstanding challenges in CML.

We recently implemented BCR::ABL1 targeted single-cell RT-qPCR in combination with index sorting for surface markers to dissect the LSC compartment at diagnosis and following 3 months of TKI therapy (*Warfvinge et al., 2017*). This demonstrated that at diagnosis, Lin⁻CD34⁺CD38⁻/low BCR::ABL1⁺ cells could be divided into seven molecularly distinct subpopulations, each reflecting unique lineage and cell state signatures. Importantly, the proportions of BCR::ABL1⁺ LSC subpopulations changed dynamically upon therapy and only a few surface markers efficiently discriminated between BCR::ABL1⁺ vs. BCR::ABL1⁻ cells. Of these, despite being generally reduced in BCR::ABL1⁺ cells upon therapy, CD26 was more frequently detected in the subpopulation most persistent during TKI treatment. The substantial treatment-induced changes observed within the stem cell population suggested that BCR::ABL1⁺ LSCs are themselves heterogeneous in terms of sensitivity to TKIs allowing us to define BCR::ABL1⁺Lin⁻CD34⁺CD38⁻/lowCD45RA⁻cKIT⁻CD26⁺ as the most TKI-insensitive cells. Apart from being detected at diagnosis, these cells are strikingly enriched after TKI therapy (*Warfvinge et al., 2017*), a finding since confirmed in a transgenic mouse model (*Shah et al., 2023*), and in clinical observations documenting the long-term persistence of CD26⁺ cells in CML patients (*Pacelli et al., 2023*). Together, these findings argue that functional heterogeneity within LSCs cannot be predicted solely by surface markers but is intimately linked to their cell state and gene expression signature, thereby motivating a strategy to simultaneously capture their lineage potency/affiliation, BCR::ABL1 status, and molecular program in addition to the surface markers.

Although recent single-cell omics studies have begun to explore the cellular landscape in CML (*Giustacchini et al., 2017*; *Krishnan et al., 2023*; *Patel et al., 2022*; *Zhang et al., 2020*), these have largely been limited to measuring single modalities at a time, and thus, multiomic investigation of BCR::ABL1⁺ LSCs and BCR::ABL1⁻ stem cells from the same patient are currently lacking. Therefore, we used cellular indexing of transcriptomes and epitopes by sequencing (CITE-seq *Stoeckius et al., 2017*) to quantify the global gene expression program, and highly multiplexed cell-surface protein

profile simultaneously within the same cells from nine CML patients. The single-cell multiomics maps enabled us to visualize the cellular heterogeneity of each patient by delineating the abundance of molecularly defined leukemic stem versus lineage progenitors, and link cell composition at diagnosis to TKI therapy response. Moreover, the maps highlighted the co-existence of molecularly distinct BCR::ABL1$^+$ vs. BCR::ABL1$^-$ stem cells, guiding their prospective isolation by novel combinations of CD26 and CD35 markers, dissection of imbued molecular programs, division kinetics and changes in heterogeneity following TKI therapy.

## Results

### Single-cell multiomic CITE-seq analysis defines the heterogeneity of CML stem cell fractions

To characterize the multimodal heterogeneity within the Lin$^-$CD34$^+$ and Lin$^-$CD34$^+$CD38$^{-/low}$ CML bone marrow (BM) compartment at diagnosis and estimate how the heterogeneity contributes to therapy response, we applied CITE-seq on nine patients retrospectively stratified according to molecular response following 12 months of treatment. To allow for accurate specification of LSPC heterogeneity without confounding effects generated from cell-type-irrelevant transcriptional signals related to BCR-ABL transformation per se, or inter-patient variability due to batch effects, we used Scarf projection (*Dhapola et al., 2022*) to map each individual CML cell to a Lin$^-$CD34$^+$ healthy bone marrow reference with a predefined heterogeneity. Each CML cell was then annotated by label transfer according to the annotation of the reference cells with the highest molecular similarity (*Figure 1A*).

The CITE-seq analysis combines single-cell RNA-seq with simultaneous staining for a panel of >40 antibodies conjugated to uniquely barcoded oligonucleotides (henceforth Antibody derived tags, ADTs) aimed to cover the spectrum of normal hematopoietic stem (CD34$^+$CD38$^{-/low}$ fraction) and lineage progenitors (CD34$^+$ enriched fraction, HSPCs) as well as surface markers reported to enrich for CML stem cells (*Herrmann et al., 2014*; *Järås et al., 2010*; *Kinstrie et al., 2020*; *Landberg et al., 2018*; *Sadovnik et al., 2016*; *Warfvinge et al., 2017*). Upon sequencing, the scRNA-seq and ADT-seq libraries provide a concurrent readout of gene expression and abundance of surface proteins, respectively. This multiomic approach surpasses scRNA-seq in cellular subtype characterization and is not constrained by the conundrum of spectral overlap of fluorophores in FACS *Stoeckius et al., 2017*, and availability of metal isotopes in mass cytometry as noted previously (*Gullaksen et al., 2019*). Our sample set of nine CML patients are from various clinical studies with attendant information on BCR::ABL1 transcript level as per the International Scale (IS %) available after 12 months of TKI therapy. These were retrospectively classified as optimal, treatment failures, and warning cases in accordance with the guidelines from European LeukemiaNet (ELN) (*Hochhaus et al., 2020 Supplementary file 1*).

To build a normal reference, we used our recently described CITE-seq profiles of Lin$^-$CD34$^+$ cells from normal BM from two age-matched healthy donors (henceforth, nBM) (*Sommarin et al., 2021*). The normal reference of 4696 Lin$^-$CD34$^+$ cells was visualized in a Uniform Manifold Approximation and Projection (UMAP, *Figure 1B*). Although single-cell omics datasets generate expression values for several thousand genes (dimensions), not all are equally informative. UMAP is a non-linear algorithm for dimensionality reduction and preserves the global data structure and enables visualization of the dataset in two dimensions. The nBM cells were subsequently clustered and annotated based on marker genes as well as information from antibody-derived tags (ADTs) against surface markers (*Figure 1C–D*, *Supplementary file 2*). We identified eleven cell clusters consisting of primitive cells, lineage progenitors and cycling cells in the UMAP, with near-to-equal distribution across the two nBM samples (*Figure 1—figure supplement 1*). The primitive cluster was marked by expression of stem-like gene expression e.g., CRHBP, HLF as well as a CD90$^+$CD38$^{-/low}$ surface profile. Closest to the primitive cluster were MPP1 and 2 without frank manifestation of lineage-specific markers. The MPP1 cluster, in turn, was connected to the clusters with gain of myeloid-like, and lymphoid-like gene expression programs. This included myelo/lymphoid (My/Ly), myeloid (My: MPO, CD33), and lymphoid/dendritic/monocyte (Ly/pDc/mono) clusters (Ly86, CD135). The MPP2 cluster, in comparison, was connected to the megakaryocyte and erythroid lineage progenitors such as megakaryocyte-erythroid (MEP), megakaryocyte (MkP: ITGA2B, CD41), erythroid progenitors (ErP: HBA1, CD71, CD105) and basophil/mast cell clusters (Baso/mc: HDC, MS4A2-3, ITGB7).

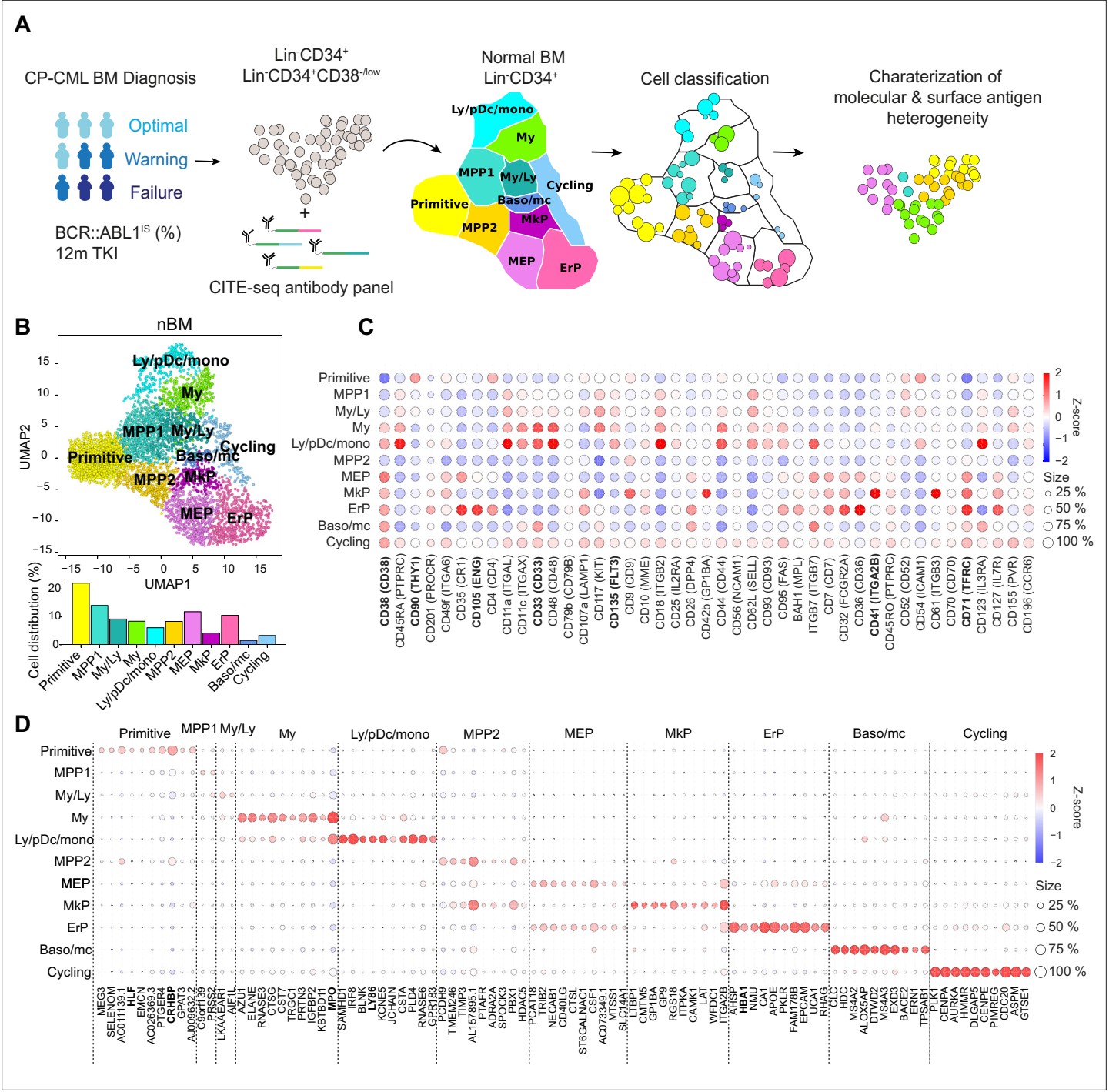

**Figure 1.** Single-cell multiomics analysis of chronic myeloid leukemia (CML) and normal bone marrow (nBM) by CITE-seq. (**A**) CML LSC and progenitor populations from nine CML bone marrow (BM) samples at diagnosis were FACS sorted and subjected to cellular indexing of transcriptomes and epitopes by sequencing (CITE-seq). CML heterogeneity was defined by label transfer through cell projection onto a reference UMAP of age-matched nBM from two donors (*Sommarin et al., 2021*). Diagnostic samples were retrospectively stratified by BCR::ABL1 International Scale (IS) % after 12 months of TKI treatment, according to the European LeukemiaNet recommendations. (**B**) UMAP embedding of 4,696 Lin⁻CD34⁺ nBM cells (n=2) with 11 clusters identified by Leiden clustering; annotated by marker genes and ADT expression (HSC = hematopoietic stem cell; MPP = multipotent progenitor population 1; My/Ly = myeloid, lymphoid progenitors; My = myeloid progenitors; Ly/pDC/mono = lymphoid, dendritic, monocytic progenitors; MPP2=multipotent progenitor population 2; MEP = megakaryocytic erythroid progenitors); MkP = megakaryocytic erythroid progenitors; ErP = erythroid progenitors; Baso/mc = basophilic, mast cell progenitors; Cycling = Cycling progenitors. The bar plot below shows the percentage of cells per cluster. (**C**) Dot plot showing the antibody-derived tags (ADT) expression within clusters. The color indicates mean ADT expression (red = high

*Figure 1 continued on next page*

*Figure 1 continued*

expression, blue = low expression); dot size represents the fraction of cells with expression. (**D**) Dot plot showing mRNA expression of the top 10 marker genes per cluster. The color indicates mean RNA expression (red = high expression, blue = low expression); dot size represents the fraction of cells with expression.

The online version of this article includes the following figure supplement(s) for figure 1:

**Figure supplement 1.** Cluster annotation by marker genes and surface protein expression.

Following CITE-seq of 14,274 Lin⁻CD34⁺ cells from nine CML patients, the cells were projected onto the nBM reference and annotated by label-transfer. Subsequently, we generated UMAPs for each patient and color-coded cells according to the transferred labels enabling us to visualize and highlight cellular heterogeneity on an individual basis (*Figure 2A*, *Figure 3—figure supplement 1*). A comparison of abundance of clusters showed that CML patients, in general, were enriched for basophil/mast cell, MEP, MkP clusters and depleted for primitive, MPP1, 2, and ly/pDC/Mono clusters (*Figure 2B*). Inspection of profiles per patient showed that while most of the clusters were consistently enriched or depleted, ErP progenitors and cycling cells showed highly individual patterns (*Figure 3—figure supplement 2B*). Furthermore, visual inspection of the individual UMAPs revealed an intriguing and distinct separation of cells within the primitive cluster in most of the patients (*Figure 2A*). Taken together, CITE-seq and label transfer through nBM reference cell projection could successfully define the molecular heterogeneity within the Lin⁻CD34⁺ LSPC at diagnosis and revealed an average increase in abundance of MEP, MkP, and basophil/mast cells as compared to nBM.

## Identification of gene signatures unique to CML cells and shared across CML stem and progenitor populations

The identification of nBM and CML primitive populations and lineage progenitors prompted us to systematically compare pair-wise gene expression differences between all eleven clusters in CML and their nBM counterparts (*Figure 2C*, *Supplementary file 5*). For each pair-wise comparison, we high-lighted only the genes that were differentially expressed between the pair but not shared with any other cluster thereby defining strict cluster-specific changes. Although this revealed unique changes across all cluster pairs, the most striking changes were seen between primitive clusters from CML versus nBM with 384 differentially expressed genes. Moreover, erythroid progenitors also displayed large changes followed by ly/pDC/mono and ErP progenitors.

In contrast, when considering genes differentially expressed in CML relative to nBM, and crucially shared among all the cluster comparisons, we identified a set of 71 genes that comprised a pan-cml signature. This approach is preferrable to a 'bulk' comparison of all CML CD34⁺ cells lumped together versus nBM cells, where the most numerically abundant cluster, and the most highly expressed genes therein, are likely to contribute disproportionately (*Figure 2D*). Of the 71 genes, 50 genes were found to be consistently up-regulated in CML stem and progenitors. Notably, the individual genes from this set were expressed at varying levels in individual clusters (*Figure 3—figure supplement 2A*, *Supplementary file 7*), reflecting a heterogenous origin of the CML signature. This included a constellation of genes such as surface markers, CD81 (*Quagliano et al., 2020*), nucleoside diphosphate kinase with transcriptional regulatory activity, NME2 (*Kar et al., 2012*; *Thakur et al., 2009*; *Thakur et al., 2014*; *Thakur et al., 2011*; *Tschiedel et al., 2012*; *Tschiedel et al., 2008*; *Yadav et al., 2014*) previously implicated in CML, genes of histone family, and components of SWI/SNF chromatin remodeling complex, BRD7 (*Kaeser et al., 2008*), as well as genes of histone methyltransferase family, KMT3A and 5 A (*Husmann and Gozani, 2019*). The analysis also identified 21 genes specifically and consistently upregulated in all clusters across nBM versus CML, and among others included DPY30 gene implicated in the maintenance of adult HSCs (*Yang et al., 2016*).

## Progenitor heterogeneity at diagnosis is linked to TKI therapy outcome

To assess how cellular heterogeneity within the Lin⁻CD34⁺ compartment at diagnosis is related to therapy response, the patients were stratified according to BCR::ABL1ᴵˢ (%) following 12 months of TKI therapy as per ELN recommendations (*Supplementary file 1*). The stem and progenitor heterogeneity at diagnosis was then evaluated relative to the treatment outcome by assessing cell composition

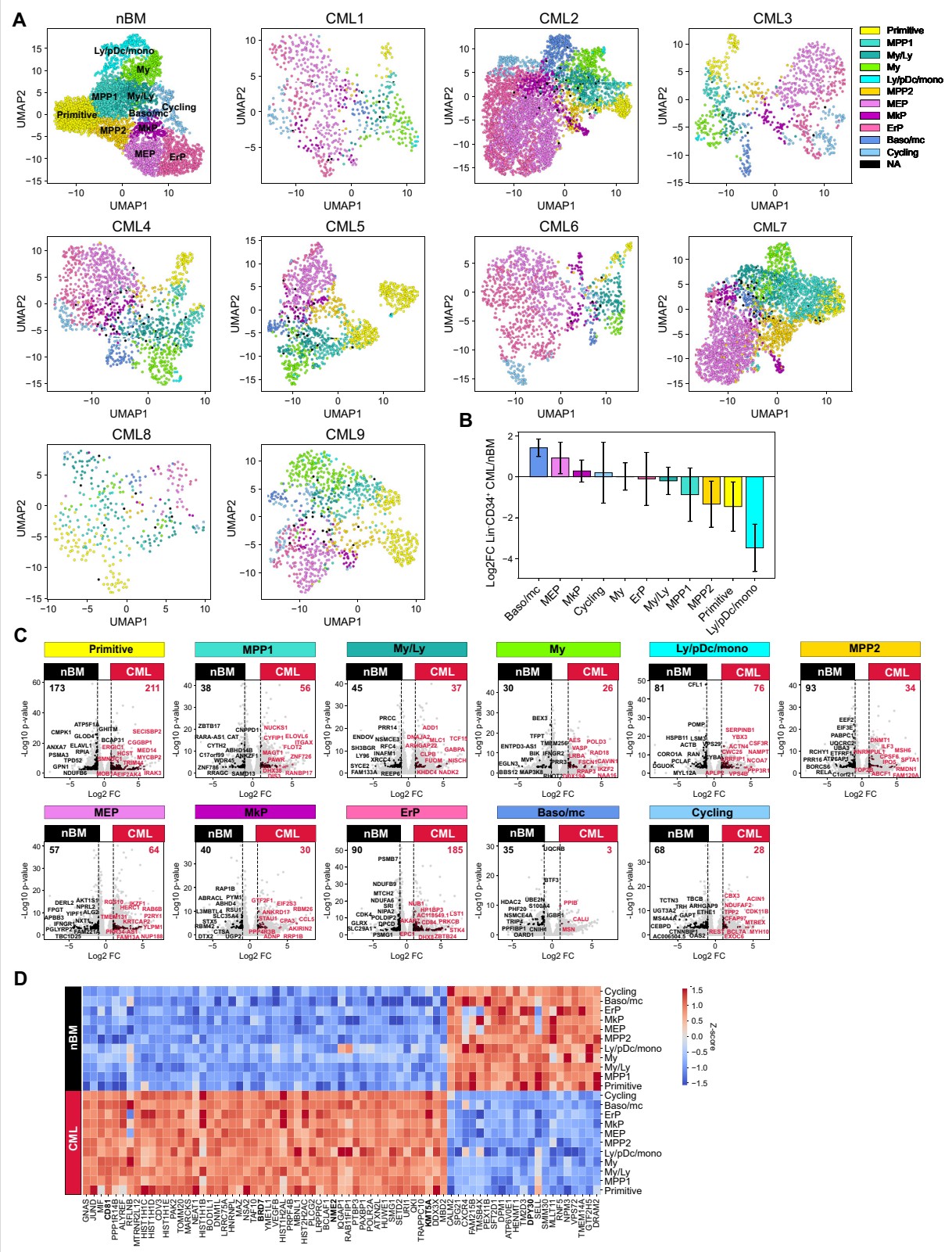

**Figure 2.** Single-cell maps of heterogeneity across patients and identification of a pan-stem and progenitor gene signature for CML. (**A**) Uniform Manifold Approximation and Projection (UMAP) embedding of Lin⁻CD34⁺ sorted chronic myeloid leukemia (CML) bone marrow (BM) cells from nine CML patients at diagnosis (14, 274 cells across patients). Cell color indicates cluster identity after label transfer through cell projection onto an aged-matched normal bone marrow (nBM) reference (the first UMAP from the left). (**B**) Bar plots showing the log2 fold change in cluster distribution (%) of Lin⁻

*Figure 2 continued on next page*

*Figure 2 continued*

CD34⁺ CML BM cells (n=9) compared to Lin⁻CD34⁺ from nBM. Error bars depict standard deviation. (**C**) Volcano plots showing differentially expressed genes between the Lin⁻CD34⁺ CML clusters (across all patients, n=9) and the corresponding clusters from nBM (adjusted p-value <.01, log2 fold change >1/< –1). Red and black dots represent genes uniquely up- or down-regulated per cluster comparison (genes not found significantly changed in any other cluster DEG analysis). The top 10 significant, unique genes are labeled in plot. Vertical dotted lines mark a log2 fold change equal to 1 and –1. (**D**) Heatmap showing the average expression of the 50 significantly up-regulated and 21 down-regulated CML signature genes (genes significantly changed and consistent through all clusters DEG analyses) across clusters from all CML patients as well as nBM (adjusted p-value <.01, log2 fold change >1/< –1). Red indicates high expression, blue low expression.

of the individual CML patients 1–9 classified as optimal (CML1-4), warning (CML 5–7), and treatment failure (CML 8–9).

By enumerating the proportion of molecularly defined clusters of individual patients, we found that all clusters from the normal bone marrow reference samples were represented in each patient. However, there was remarkable heterogeneity between patients in terms of relative abundance of constituent clusters (*Figure 3A*, *Figure 3—figure supplement 1B*). Importantly, this inter-patient heterogeneity followed patterns connected to therapy outcome. A focused comparison between optimal responders (CML 1–4) and treatment failure cases (CML 8–9) showed distinctive enrichment of specific cell clusters (*Figure 3B*), where failure cases displayed a higher abundance of the molecularly defined primitive cluster, multipotent progenitors 1 and 2, and myelo-lymphoid (My/Ly) and Ly/pDC/monocyte clusters at diagnosis. In contrast, the optimal responders had a higher burden of megakaryocyte-erythroid progenitors (MEP), megakaryocyte progenitors (MkP), Erythroid progenitor (ErP), Baso/mc, and cycling clusters at diagnosis. Among others, the most conspicuous changes between optimal responders and treatment failures were observed in the primitive cluster (fourfold higher in failures), and MEP and erythroid cluster (~threefold higher in optimal). The warning cases (CML 5–7) also showed heterogeneity in terms of cell composition (*Figure 3A*); for instance, while CML 5 had a relatively higher content of primitive cells akin to treatment failures, CML 6 and 7 had a profile resembling optimal responders. Thus, these observations indicate that the prospective treatment outcome is reflected by the composition of stem and progenitor cell types at diagnosis.

## Large-scale deconvolution of an independent dataset shows that cell composition is linked to cytogenetic response to TKI therapy

To further assess whether the hematopoietic stem and progenitor composition of CML patients at diagnosis is connected to their therapy response, we utilized an independent publicly available bulk gene expression dataset from CML patients with known therapy response (n=59 *McWeeney et al., 2010*). This was done with an aim to computationally deconvolute the individual profiles into constituent cell types by employing the same nBM used to define the heterogeneity in the CITE-seq data as a molecular reference, subsequently estimate their relative abundances, and relate to TKI response (*Figure 3C*). We used CIBERSORTx (*Newman et al., 2019*), a widely used algorithm, which has also been recently used to deconvolute bulk transcriptomes from acute myeloid leukemia (AML) (*Zeng et al., 2022*). As the gene expression profiles were measured at diagnosis from CD34⁺ enriched cells from either bone marrow or peripheral blood, this represented a cellular fraction broadly similar to ours for comparison. Importantly, the CML patient cohort was stratified by the percentage of Ph⁺ cells after 12 months of Imatinib treatment as responders (0% Ph⁺ metaphases, complete cytogenetic response, CcyR) or non-responders (>65% Ph⁺ metaphases, lack of even a minor cytogenetic response) as per the original study (*McWeeney et al., 2010*). We adhered to the patient annotation (responder or non-responder) provided by the original authors for consistency (see methods for more details).

The deconvolution results demonstrated a heterogeneous cell composition across patients, and overall comparison of imatinib non-responders (n=18 out of 59) versus responders (n=41 out of 59) showed a >threefold statistically significant enrichment of primitive cells in non-responders (*Figure 3D*, *Figure 3—figure supplement 2B*). To assess whether the source of origin of CD34⁺ cells in the public dataset could bias our results, we compared the abundances of imputed cell types from non-responders and responders based on their extraction from either bone marrow, or peripheral blood separately. This once again revealed a statistically enriched presence of primitive cells in non-responders versus the responders in the bone marrow (~threefold higher). Moreover, and unlike the

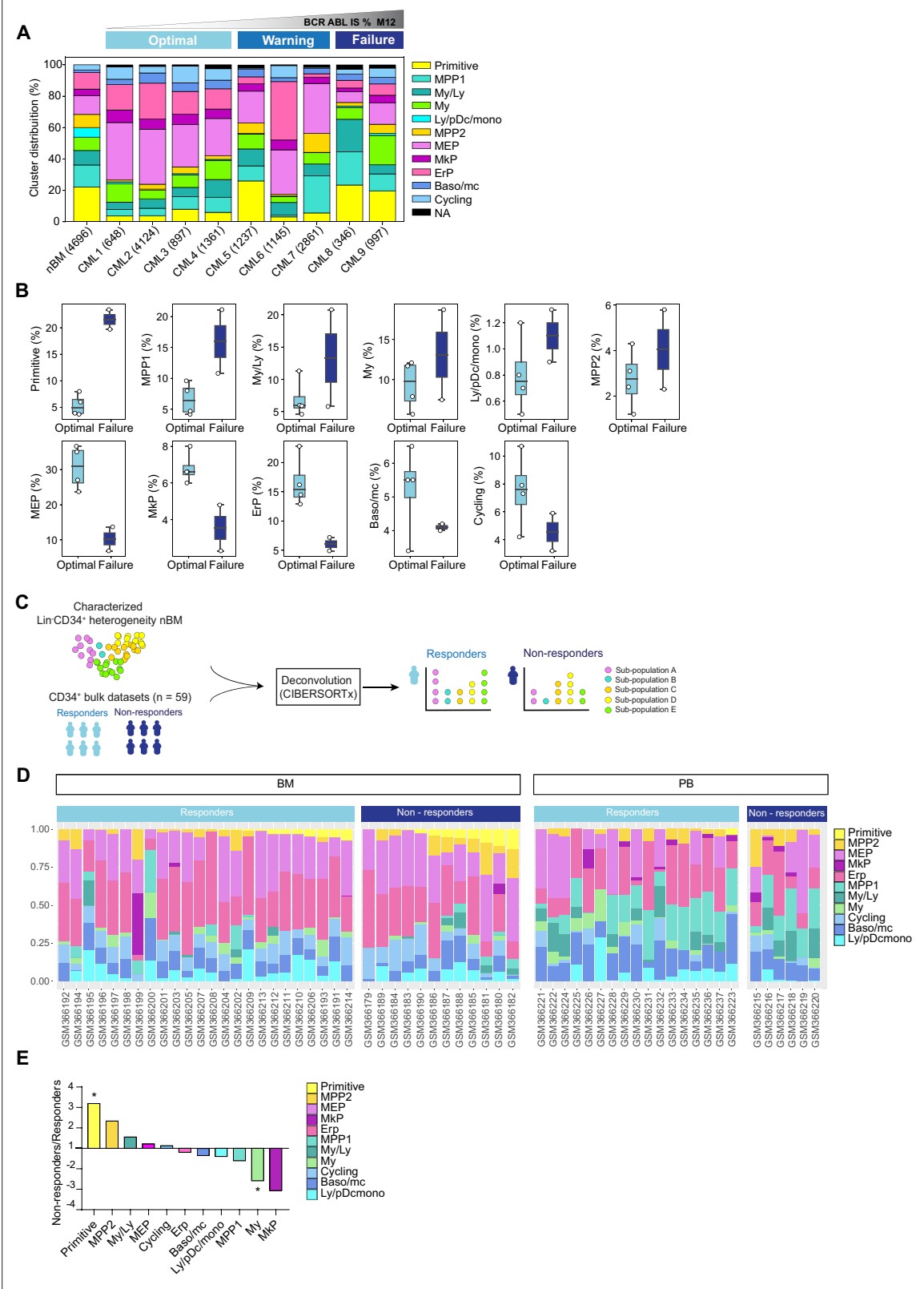

**Figure 3.** Cell heterogeneity at diagnosis relates to tyrosine kinase inhibitor (TKI) therapy outcome in chronic myeloid leukemia (CML). (**A**) Stacked bar plots showing the cluster distribution within Lin⁻CD34⁺ compartment in all CML patients at diagnosis as well as normal bone marrow (nBM). CML patients are ordered by BCR::ABL1^IS (%) following 12 months of TKI therapy (M12) and retrospectively stratified as per European LeukemiaNet recommendations; Optimal = CML1-4 (≤0.1%), Warning = CML5-7 (>0.1–1%), Failure = CML8-9 (>1 %). The total number of Lin⁻CD34⁺ cells from

*Figure 3 continued on next page*

Figure 3 continued

individual patients is indicated at the bottom. (**B**) Box plots comparing cluster proportions in optimal responders (n=4, CML patients 1–4) and treatment failures (n=2, CML patients 8–9) at diagnosis; the patients were retrospectively stratified according to BCR::ABL1$^{IS}$ (%) after 12 months of TKI therapy (Optimal ≤1 %, Failure >1 %). (**C**) A scheme for computational deconvolution of bulk transcriptomes from patients into constituent cell populations; using cellular indexing of transcriptomes and epitopes by sequencing (CITE-seq) derived gene signatures from Lin⁻CD34⁺ nBM as reference, an independent bulk CD34⁺ microarray dataset from CML patients (n=59 **McWeeney et al., 2010**) was deconvoluted into constituent cell populations using CIBERSORTx (**Newman et al., 2019**). (**D**) Stacked bar plots showing the percentage of specific clusters within CD34⁺ cells from individual CML patients (n=59). The x-axis shows individual GSE ID for patients, y-axis shows the percentage of clusters with similar color codes as used in **Figures 1–2** UMAPs. Annotation of individual patients as per the original study (**McWeeney et al., 2010**); non-responders with >65% Ph + metaphases (did not achieve even a minor cytogenetic response), responders with 0% Ph + metaphases after 12 months of Imatinib therapy (achieved CCyR). BM and PB represent CD34⁺ cells isolated from bone marrow, and peripheral blood, respectively. (**E**) The fold change in proportions for cell clusters between non-responders and responders (annotation as described above) for CD34⁺ cells isolated from bone marrow; statistical significance shown by asterisk *; student t-test, p-value <0.05.

The online version of this article includes the following figure supplement(s) for figure 3:

**Figure supplement 1.** Heterogeneity of CML patient HSPCs at diagnosis.

**Figure supplement 2.** Molecular chararacterization of CML HSPC clusters.

findings from overall comparison, myeloid cells were also found to be statistically enriched within responders (~2.5 fold higher **Figure 3E**). Although MPP2 and MkP clusters showed increased abundance in non-responders and responders, respectively, the difference did not reach statistical significance. Notably, the primitive cells were rarely detected in peripheral blood in contrast to the bone marrow, confirming the notion that cell type fractions differ in different tissues (**Figure 3D**), and the bone marrow is predictably a more reliable and abundant source of true primitive cells in CML. Taken together, our analyses revealed distinct cellular heterogeneity across CML patients at diagnosis and highlighted that cell composition is associated with therapy response. Specifically, a higher burden of cells with a stem-like signature at diagnosis is a strong indicator of treatment failure.

## Identification of BCR::ABL1⁺ and BCR::ABL1⁻ primitive cells and their surface markers by combining single-cell gene expression with ADTs

Interestingly, according to the ADT data, the established CML stem cell markers CD25, CD26, and CD93 could only capture a fraction of the primitive cells (**Figure 3—figure supplement 2C**). To further investigate the molecular and immunophenotypic heterogeneity within the primitive cluster we performed CITE-seq on the stem cell enriched Lin⁻CD34⁺CD38⁻/low compartment from the same patients (n=8; 6779 cells **Figure 4—figure supplement 1A** and **Figure 4—figure supplement 2A**). A stepwise analysis (**Figure 4A**) was performed to resolve the BCR::ABL1 status of the individual primitive cells and determine their immunophenotype. First (1), the primitive cells from the Lin⁻CD34⁺CD38⁻/low fraction of all CML patients were merged and classified by BCR::ABL1 status using publicly available BCR::ABL1⁺ gene signatures. Next (2) CITE-seq data from both the Lin⁻CD34⁺ and the Lin⁻CD34⁺CD38⁻/low sorted population from the same patient was merged and visualized in a joint UMAP, while (3) the BCR::ABL1 status of the Lin⁻CD34⁺CD38⁻/low primitive cells was linked to the merged UMAPs by cell ID. Finally (4) the log2 fold change in ADT expression between BCR::ABL1⁺ and BCR::ABL1⁻ primitive cells were calculated and the remaining cells of the UMAP were color-coded by the cluster identity generated by label-transferred projection to nBM.

The BCR::ABL1 status of the primitive cells was determined using two previously established gene expression signatures comparing Lin⁻CD34⁺CD38⁻/low BCR::ABL1⁺ LSCs to normal Lin⁻CD34⁺CD38⁻/low HSCs, and Lin⁻CD34⁺CD38⁻/low BCR::ABL1⁻ non-leukemic stem cells (**Giustacchini et al., 2017**). These signatures have been derived by analyzing cells with SMART-seq for global transcriptome, and primers specific to the BCR::ABL1 fusion gene, simultaneously and, therefore, represent useful anchors to query and label cells as either BCR::ABL1⁺ or BCR::ABL1⁻ (See methods for detailed analysis).

In the UMAP generated from all Lin⁻CD34⁺CD38⁻/low primitive cells, we identified three coarse clusters, where two consisted of primitive CML cells with representation from all patients (cluster 2–3), and one represented Lin⁻CD34⁺CD38⁻/low primitive cells from nBM which were included as negative control (cluster 1 **Figure 4—figure supplement 1B–C**). When we queried each cluster for enrichment of either LSC or HSC-like gene signatures the analysis clearly demonstrated that the primitive cells from CML samples could be divided based on the presence of BCR:ABL1, where CML cluster 3

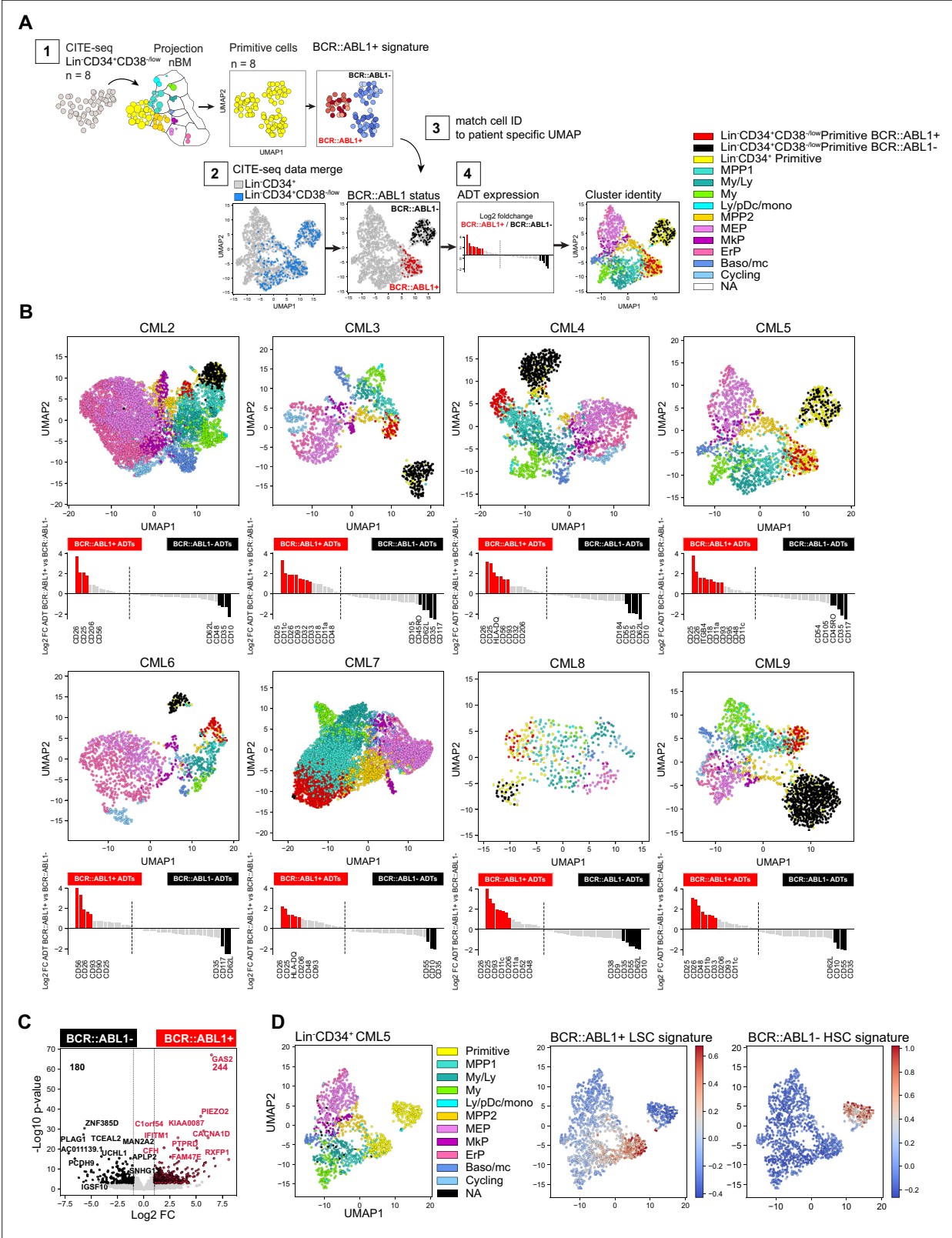

**Figure 4.** Identification of BCR::ABL1+ and BCR::ABL1- primitive cells and their surface markers by joint analysis of single-cell gene expression and multiplexed antibody-derived sequence tags (ADTs). (**A**) Overview of analysis steps to generate (**B**) (1) Lin⁻CD34⁺CD38⁻/low cells were projected onto Lin⁻CD34⁺ nBM. Subsequently, the primitive cluster CITE-seq data from all CML patients and nBM (negative control) were merged and visualized together in one Uniform Manifold Approximation and Projection (UMAP). BCR::ABL1⁺ gene signatures from BCR::ABL1 targeted SMART-seq (*Giustacchini et al.,*

*Figure 4 continued on next page*

*Figure 4 continued*

*2017*) was used to define primitive cluster cells from the individual patients as either BCR::ABL1$^+$ or BCR::ABL1$^-$ (2) Lin$^-$CD34$^+$ and Lin$^-$CD34$^+$CD38$^{-/low}$ CITE-seq data from the same patient was merged and visualized together in a UMAP (3) BCR-ABL status from the Lin$^-$CD34$^+$CD38$^{-/low}$ primitive cells was linked to the joint UMAPs by matching cell IDs and cell were colored as either red (BCR::ABL1$^+$) or black (BCR::ABL1$^-$) (4) The log2 fold change in antibody-derived tags ADT expression between primitive Lin$^-$CD34$^+$CD38$^{-/low}$ BCR::ABL1$^+$ (red) and BCR::ABL1$^-$ (black) cells was calculated and the remaining cells were colored according to their cluster annotation given after projection onto Lin$^-$CD34$^+$ nBM. (**B**) UMAP plots showing the merged CITE-seq data of Lin$^-$CD34$^+$ and Lin$^-$CD34$^+$CD38$^{-/low}$ sorted populations for CML2-9 individually. Cells are color-coded according to cluster annotation given following projection onto Lin$^-$CD34$^+$ nBM. Lin$^-$CD34$^+$CD38$^{-/low}$ primitive cluster cells are annotated as BCR::ABL1$^+$ (red) or BCR::ABL1$^-$ (black) and display enrichment of BCR::ABL1$^+$ LSC signatures and non-leukemic stem cell signatures, respectively. The red and black bars in the bar plot below indicate ADTs with significant changes in expression (p-value <.05, log2 fold change >1 / < –1) for BCR::ABL1$^+$ and BCR::ABL1$^-$ cells, respectively. (**C**) Volcano plot showing differentially expressed genes between CML Lin$^-$CD34$^+$CD38$^{-/low}$ Primitive BCR::ABL1$^+$ and BCR::ABL1$^-$ cells. Red and black dots represent significant up- and down-regulated genes (adjusted p-value <.01, log2 fold change >1 / < –1). The top 10 significant genes are labeled. Vertical dotted lines mark a fold change equal to 1 and –1. (**D**) UMAP embedding of CML Lin$^-$CD34$^+$ cells from one representative CML patient at diagnosis (CML5). First plot from the left shows cells colored according to the cluster identity given after mapping to nBM, the primitive cells are colored yellow. The following two UMAPs of Lin$^-$CD34$^+$ show the relative mRNA expression of the BCR::ABL1$^+$ LSC signature and BCR::ABL1$^-$ non-leukemic stem cell gene signature; scale: red = high expression, blue = low expression.

The online version of this article includes the following figure supplement(s) for figure 4:

**Figure supplement 1.** Identification of BCR::ABL+ stem cells.

**Figure supplement 2.** Mapping of BCR::ABL+ stem cells.

**Figure supplement 3.** Mapping of BCR::ABL+ stem cells in individual patients.

**Figure supplement 4.** Mapping of BCR::ABL+ stem cells in individual patients.

---

displayed an evident enrichment of BCR::ABL1$^+$ LSC specific signatures as compared to CML cluster 2 consisting of BCR::ABL1$^-$ non-leukemic stem cells (*Figure 4—figure supplement 1D–E*). Interestingly, both CML clusters were also found to share gene characteristics with normal stem cells (*Figure 4— figure supplement 1D*). Accordingly, we annotated Lin$^-$CD34$^+$CD38$^{-/low}$ primitive cells from individual patients as either BCR::ABL1$^+$ or BCR::ABL1$^-$ and could observe that the distribution varied across patients, suggesting a variable load of leukemic stem and normal stem-like cells within the primitive fraction (*Figure 4—figure supplement 1F*, *Figure 4—figure supplement 2A*).

Remarkably, subsequent matching of BCR::ABL1$^+$ and BCR::ABL1$^-$ cell IDs to the primitive cells in the joint Lin$^-$CD34$^+$ and Lin$^-$CD34$^+$CD38$^{-/low}$ UMAPs (indicated by red and black cell color, respectively), indeed revealed that BCR::ABL1$^+$ LSCs and BCR::ABL1$^-$ HSCs were distinctly and consistently separated in the UMAP for each patient (*Figure 4B*, *Figure 4—figure supplement 3*). BCR::ABL1$^+$ LSCs in all patients were positioned immediately juxtaposing the downstream progenitor populations forming a continuum of differentiation hierarchy characteristically observed in single-cell analysis of HSPCs (*Pellin et al., 2019*; *Safi et al., 2022*; *Velten et al., 2017*). In contrast, BCR::ABL1$^-$ HSCs formed an isolated and distinct cluster clearly detached from the rest of the cells in the majority of patients, indicating that during the chronic phase of CML, active hematopoiesis is dominated by the BCR::ABL1$^+$ LSCs while BCR::ABL1$^-$ HSCs reside in the bone marrow albeit in an inactive state with reduced contribution to hematopoiesis as previously suggested (*Chen et al., 2023*; *Coulombel et al., 1983*). Moreover, using the ADT data we could validate the molecularly defined BCR::ABL1$^+$ LSCs also by surface expression, with CD26, CD25, and CD93 displaying a 2–8 fold elevated expression in BCR::ABL1$^+$ primitive cells as compared to BCR::ABL1$^-$. A noteworthy patient-specific variability in surface protein expression on the most primitive cells was also established (*Figure 4B*).

To gain an understanding of the nature of molecular program of BCR::ABL1$^+$ vs. BCR::ABL1$^-$ primitive cells, we analyzed differentially expressed genes between the pooled primitive BCR::ABL1$^+$ and BCR::ABL1$^-$ populations from all CML patients. This identified 244 genes specifically upregulated in BCR::ABL1$^+$ and 180 genes upregulated in BCR::ABL1$^-$ cells (*Figure 4C*, *Supplementary file 6*). Interestingly, among genes up-regulated in BCR::ABL1$^+$ primitive cells included the expression of established CML markers; IL2RA (CD25), DPP4 (CD26), and IL1RAP as well as LEPR (CD295) a receptor recently described as a novel marker for Lin$^-$CD34$^+$CD38$^{-/low}$ LSCs (*Landberg et al., 2018*). In contrast, BCR::ABL1$^-$ cells displayed an up-regulated expression of the stem cell-related gene CRHBP (*Barbieri et al., 2022*; *He et al., 2005*) as well as CXCR4 important for homing and maintenance of normal HSCs (*Sugiyama et al., 2006*). Upon applying the gene signatures to Lin$^-$CD34$^+$ cells, we could clearly identify as well as partition primitive cells into BCR::ABL1$^+$ vs. BCR::ABL1$^-$ (*Figure 4D* and

*Figure 4—figure supplement 3*). This is especially relevant as the Lin⁻CD34⁺ fraction contains a higher proportion of lineage progenitors but a smaller fraction of primitive cells making their identification challenging. Taken together, this multistep analysis approach generated unique detailed UMAPs allowing for inspection of BCR::ABL1⁺ LSCs and BCR::ABL1⁻ HSCs, their relation to other progenitor cells, as well as further validate BCR::ABL1⁺ LSCs identity by surface protein expression.

## A unique combination of CD26 and CD35 surface markers captures molecularly defined BCR::ABL1⁺ LSCs and BCR::ABL1⁻ HSCs

A long-sought aim in CML research has been to generate protocols for effective discrimination and isolation of CML LSCs from residual HSCs within individual patient bone marrow. The single-cell multiomic feature space of CITE-seq together with the identification of molecular distinct BCR::ABL1⁺ LSCs and BCR::ABL1⁻ HSCs provides a unique opportunity to identify efficient protocols for their separation.

To specifically define the surface markers for their ability to capture BCR::ABL1⁺ vs. BCR::ABL1⁻ primitive cells, we compared significantly up-regulated ADTs for either Lin⁻CD34⁺CD38⁻/low BCR::ABL1⁺ LSCs or BCR::ABL1⁻ HSCs cells across patients and identified markers with consistent expression across the data set. We found 28 unique markers to be significantly up-regulated for either BCR::ABL1⁺ or BCR::ABL1⁻ cells (black or red colored bars in *Figure 4B*, *Figure 5A*). Of these 28 markers, 17 ADTs marked BCR::ABL1⁺ LSCs and were detected in at least one patient. However, only CD26, and CD25 were explicit and consistent throughout the entire cohort. CD93 was present on BCR::ABL1⁺ LSCs in 7/8 patients and other previously suggested LSC markers like CD33 and CD56 specifically labeled BCR::ABL1⁺ primitive cells in only a fraction of the patients. The documented CML LSC marker IL1RAP was not in the ADT panel and hence could not be probed. In contrast, 11 ADTs were specifically detected in BCR::ABL1⁻ primitive cells from at least one patient. Interestingly, CD35 with ~fourfold elevated expression as compared to BCR::ABL1⁺ LSCs was the only marker found to be consistently specific for the BCR::ABL1⁻ HSC population across all patients. CD62L and CD10 were specifically present on BCR::ABL1⁻ HSCs in 6/8 and 5/8 patients, respectively. Intriguingly, we recently showed that CD35 captures all HSCs in healthy human hematopoiesis and that stem cell activity as measured by xenotransplantation assays and epigenetic profiling is restricted to CD35⁺ HSCs (*Sommarin et al., 2021*). By applying the BCR::ABL1 signatures also to the Lin⁻CD34⁺ primitive cells (*Figure 4—figure supplement 4*), we could visualize the relative expression within the Lin⁻CD34⁺ UMAPs in relation to the cells BCR::ABL1 status and further confirm that the primitive fraction, previously captured by the established CML stem cell markers CD25, CD26, and CD93, consisted of specifically BCR::ABL1⁺ LSCs (*Figure 5B–C*, *Figure 5—figure supplement 1* and *Figure 5—figure supplement 2A*). In addition, we observed that our previously described surface marker CD117, whose absence characterized LSCs persisting 3 months of TKI therapy, displayed a highly variably expression at diagnosis both marking a proportion of LSCs and HSCs across patients. Interestingly, despite that relative expression of CD35⁺ was lower in the primitive cells compared to MEP/ERP cell clusters the CD35⁺ gate captured ~50% of the BCR::ABL1⁻ primitive cluster.

Given the consistency of detection across patients, we specifically focused on the ability of ADTs against CD25, CD26, and CD35 to separate between BCR::ABL1⁺ LSCs and BCR::ABL1⁻ HSCs within the stem cell-enriched Lin⁻CD34⁺CD38⁻/low compartment. CD26 showed the highest capture of the BCR::ABL1⁺ primitive cluster (>95%) with a consistent capture percentage across patients (*Figure 5D*). This was followed by CD25 which captured >80% of the BCR::ABL1⁺ primitive cluster on average but displayed higher variability in capture efficiency compared to CD26 across patients. In comparison, the percentage of BCR::ABL1⁻ primitive cluster captured by CD26 and CD25 was considerably lower (~5%). In contrast, the CD35 ⁺ gate showed a remarkable specific capture of BCR::ABL1⁻ primitive cluster (~50%) (*Figure 5D*).

To further confirm whether CD26 and CD35 together could efficiently separate BCR::ABL1⁺ and BCR::ABL1⁻ primitive cells, we gated for different combinations of CD26 and CD35 within the Lin⁻CD34⁺CD38⁻/low compartment (*Figure 5E*, *Figure 5—figure supplement 2B*). Predictably, whereas a combination of CD26⁺CD35⁻ ADTs captured BCR::ABL1⁺ primitive LSC cluster at high purity, the CD26⁻CD35⁺ combination marked BCR::ABL1⁻ primitive HSC cluster with no contamination of BCR::ABL1⁺ LSCs (*Figure 5E*). Interestingly, CD35 expression divided the CD26⁻, BCR::ABL1⁻ primitive cells in one CD35 +and one CD35- fraction. This is in line with our recent observation that human

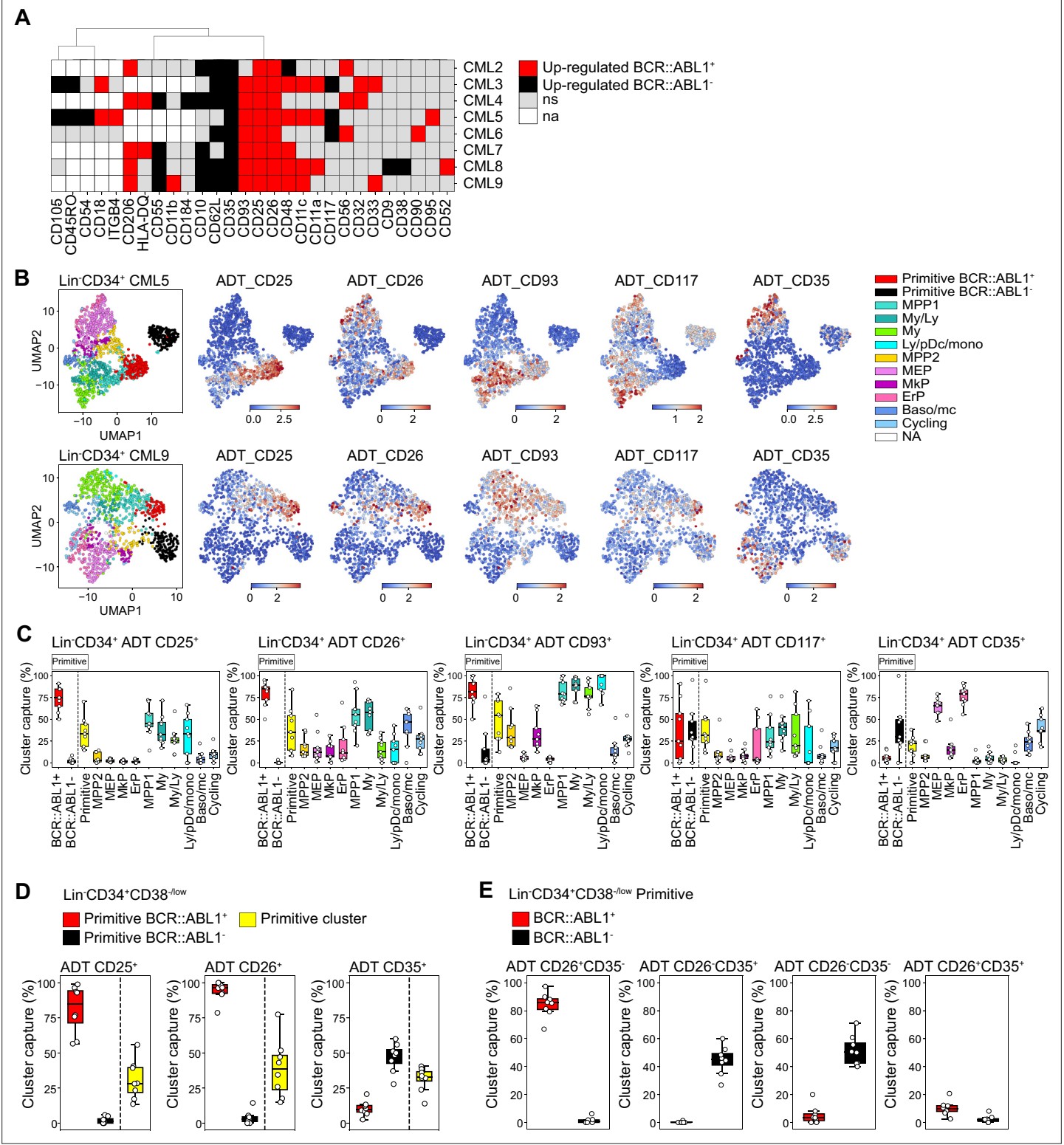

**Figure 5.** Assessment of surface marker combinations to capture molecularly defined BCR::ABL1+ and BCR::ABL1- primitive cells. (**A**) Heatmap comparing antibody-derived tags (ADT) expression between BCR::ABL1+ and BCR::ABL1- cells within the Lin-CD34+CD38-/low primitive cluster across chronic myeloid leukemia (CML) patients (black = surface markers significantly up-regulated in BCR::ABL1- cells (log2 fold change < –1, p-value <.05); red = surface markers significantly up-regulated in BCR::ABL1+ cells (log2 fold change >1, p-value <.05); gray = non-significant change in ADT expression; white = surface marker was not present in the ADT panel used for the specific patient). (**B**) Visualization of the relative ADT expression of CD25, CD26, CD93, CD117, and CD35 in the UMAPs of Lin-CD34+ cells from CML patients 5 and 9 (red = high expression, blue = low expression).

*Figure 5 continued on next page*

Figure 5 continued

(C) Assessment of a selection of ADTs; CD25, CD26, CD93, CD117, and CD35 capacity to capture BCR::ABL1$^+$ primitive, BCR::ABL1$^-$ primitive, all primitive cells, and specific progenitors across patients by gating on their expression within the Lin$^-$CD34$^+$ compartment (x-axis shows specific clusters, y-axis: percentage of specific cluster captured; each white circle represents a patient sample). (D) Assessment of specific ADTs: CD26, CD25, and CD35 capacity to capture BCR::ABL1$^+$ primitive versus BCR::ABL1$^-$ primitive cluster within the Lin$^-$CD34$^+$CD38$^{-/low}$ compartment across patients; x-axis shows specific clusters, y-axis: percentage of specific cluster captured; each white circle represents a patient sample. (E) Assessment of specific combinations of ADTs: CD26$^+$CD35$^-$, CD26$^-$CD35$^+$, CD26$^-$CD35$^-$, and CD26$^+$CD35$^+$ capacity to capture BCR::ABL1$^+$ primitive versus BCR::ABL1$^-$ primitive cluster within the Lin$^-$CD34$^+$CD38$^{-/low}$ compartment across patients; x-axis shows specific clusters, y-axis: percentage of specific cluster captured; each white circle represents a patient sample.

The online version of this article includes the following figure supplement(s) for figure 5:

**Figure supplement 1.** Expression of ADTs across pateints.

**Figure supplement 2.** Validation of CD26 and CD35 stem cell populations.

BM immunophenotypic stem cells can be divided by CD35, where all HSC activity is captured within the CD35$^+$ fraction (*Sommarin et al., 2021*). Accordingly, ADT combination of CD26$^-$CD35$^-$ captured BCR::ABL1$^-$ primitive cells likely representing non-leukemic MPPs downstream of normal CD26$^-$CD35$^+$ stem cells. Notably, only a minority of cells were positive for ADTs for the CD26$^+$CD35$^+$ combination as evidenced by an overall low capture of cells in general. Thus, this shows the importance of positive, HSC-specific markers for efficient isolation of CML LSCs, and detection of residual HSCs in leukemia. Taken together, these observations suggested that although patients are heterogeneous for a given marker, BCR::ABL1$^+$ LSCs can be consistently captured by the CD26$^+$CD35$^-$ combination while BCR::ABL1$^-$ HSCs expressed a CD26$^-$CD35$^+$ immunophenotype within Lin$^-$CD34$^+$CD38$^{-/low}$ compartment in the bone marrow of CML patients.

## A comparative analysis of CD26$^+$CD35$^-$ versus CD26$^-$CD35$^+$ cells for BCR::ABL1 expression and response to TKI therapy

To explore whether the combination of CD26 and CD35 could indeed purify BCR::ABL1$^+$ vs. BCR::ABL1$^-$ in Flow Cytometry protocols, we sorted CD26$^+$CD35$^-$ and CD26$^-$CD35$^+$ cells from the Lin$^-$CD34$^+$CD38$^{-/low}$ compartment from patients both before and following 3 months of Bosutinib therapy and employed quantitative real-time qPCR analysis using Taqman probes against the BCR::ABL1 fusion gene (*Figure 6A*). The analysis showed a strong signal within CD26$^+$CD35$^-$ cells for BCR::ABL1 expression while CD26$^-$CD35$^+$ cells were BCR::ABL1$^{low/negative}$, validating the capacity of the combination of CD26 and CD35 to separate BCR::ABL1$^+$ and BCR::ABL1$^-$ stem cells at diagnosis as well as after second generation TKI therapy. In addition, real-time qPCR analysis confirmed that Lin$^-$CD34$^+$CD38$^{-/low}$CD26$^+$CD35$^-$ cells from CML patients at diagnosis displayed a strong signal for GAS2 (*Figure 6B*), one of the top genes from the CITE-seq BCR::ABL1$^+$ LSC signature (*Figure 4C*). Furthermore, CD26$^+$CD35$^-$ cells maintained GAS2 expression following 3 months of Bosutinib therapy (*Figure 6B*). GAS2 has previously been linked to CML disease progression, cell cycle control, apoptosis, and response to Imatinib (*Janssen et al., 2005*; *Radich et al., 2006*; *Zhou et al., 2014*).

To assess the level of quiescence between the BCR::ABL1$^+$ LSCs and BCR::ABL1$^-$ HSCs, we sorted Lin$^-$CD34$^+$CD38$^{-/low}$CD45RA$^-$CD26$^+$CD35$^-$ and CD26$^-$CD35$^+$ single cells from two CML patients at diagnosis into individual wells and examined their division kinetics for 140 hr. However, LSC and HSC populations were indistinguishably deeply quiescent with average times to first division of 80–100 hr in culture (*Figure 5—figure supplement 2D–C*). Moreover, cell cycle analysis was performed by gating the corresponding cell populations in the Lin$^-$CD34$^+$CD38$^{-/low}$ CITE-seq data using ADTs. In accordance with the in vitro study, there was no distinct difference in cell cycle status between the populations (*Figure 5—figure supplement 2E–F*).

The higher burden of primitive cells in treatment failures observed above prompted us to query to which extent patients differed in the load of isolatable BCR::ABL1$^+$ vs. BCR::ABL1$^-$ at diagnosis and their sustenance in response to TKI therapy. Intriguingly, we noted a substantially higher ratio of CD26$^+$CD35$^-$ LSCs over CD26$^-$CD35$^+$ HSCs at diagnosis in prospective treatment failures compared to the bone marrow of optimal responders, both using CITE-seq ADTs as well as in FACS analysis for CD26 and CD35 (*Figure 6C–D*, *Figure 5—figure supplement 2B*). Importantly, following three months of TKI therapy, the percentage of BCR::ABL1$^+$ CD26$^+$CD35$^-$ cells decreased in both groups of patients; however, there was a striking difference in the relative proportion of CD26$^+$CD35$^-$ LSCs/CD26$^-$CD35$^+$

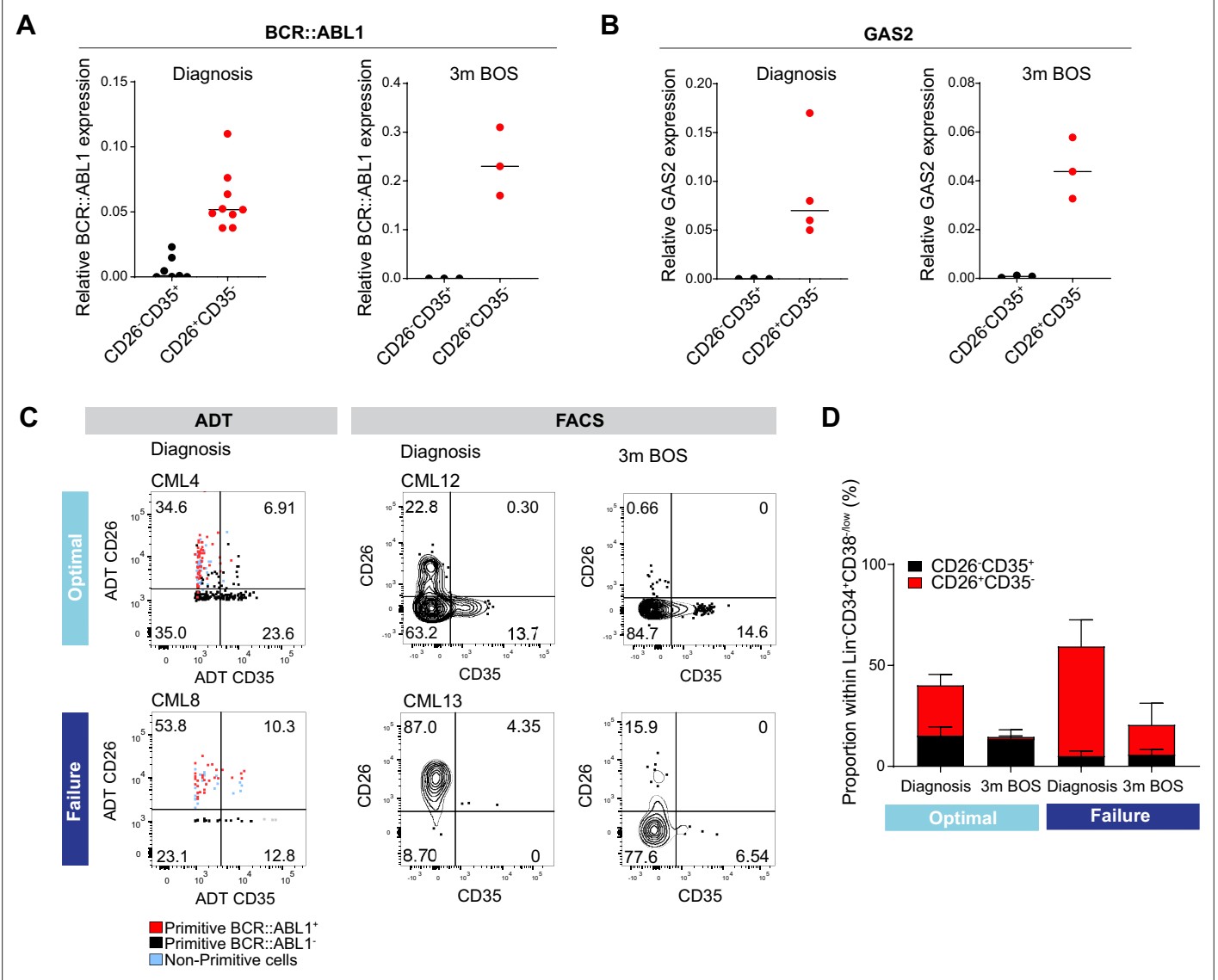

**Figure 6.** Analysis of CD26+CD35- vs. CD26-CD35+ cells for BCR::ABL1 transcript expression, and response to tyrosine kinase inhibitor (TKI) therapy. (**A**) Real-time qPCR for BCR::ABL1 in CD26-CD35+ (n=7, CML patients 10–12, 14–15, 17–18) vs. CD26+CD35- (n=9, chronic myeloid leukemia (CML) patients 10–15, 17–18, and 20) within the Lin-CD34+CD38-/low compartment at diagnosis and following 3 months of Bosutinib therapy (n=3, CML patients 17, 18, 21). GAPDH served as control. (**B**) Real-time qPCR for GAS2 in CD26-CD35+ (n=3, CML patients 14, 17–18) vs. CD26+CD35- (n=4, CML patients 13–14, 17–18) within the Lin-CD34+CD38-/low compartment at diagnosis and following 3 months of Bosutinib therapy (n=3, CML patients 17–18, 21). GAPDH served as control. (**C**) Representative FACS plots showing the percentage CD26+CD35- (leukemic stem cells, LSC) and CD26-CD35+ (hematopoietic stem cell, HSC) cells within the Lin-CD34+CD38-/low compartment. Left panel: antibody-derived tags (ADT)-gated CD26+CD35- and CD26-CD35+ cells within Lin-CD34+CD38-/low CITE-seq data from selected optimal responder (CML4) and treatment failure (CML8) at diagnosis. Right panel: FACS gated Lin-CD34+CD38-/lowCD26+CD35- and CD26-CD35+ cells in selected optimal responder (CML12) and treatment failure (CML13) at diagnosis versus 3 months of TKI therapy (Bosutinib). (**D**) Bar plots showing the percentage of CD26+CD35- and CD26-CD35+ cells within Lin-CD34+CD38-/low compartment from optimal responders and treatment failures at diagnosis n=11; optimal = 6 (≤0.1%, CML patients 12, 14–18), failure = 5 (>1%, CML patients 13, 19, 22–23, 25) and following 3 months of Bosutinib therapy n=11; optimal = 6 (≤0.1%, CML patients 12, 14–18), failure = 5 (>1%, CML patients 13, 19, 22–23, 24) determined using FACS.

HSCs in both groups. The stem cell compartment of optimal responders was dominated by healthy HSCs at three months following treatment. In contrast, the stem cell population of failure patients still comprised of CD26+CD35- LSCs without re-establishment of HSCs and restoration of normal hematopoiesis (*Figure 6E*). Taken together, we here for the first time present a protocol for separation and

isolation of both BCR::ABL1+ LSCs and BCR::ABL1- HSCs from the same CML patients, two populations evidently relevant to therapy response.

## Discussion

As new and more potent TKIs continue to be developed (*Braun et al., 2020*), the persistence of fully leukemogenic cells even in TFR necessitates life-long therapy in most patients. The ensuing treatment-emergent adverse events, and financial toxicity (*Cortes et al., 2021*; *Lipton et al., 2022*; *Zafar, 2016*) motivate a search for organizing principles to predict therapy response, dampen disease progression, and achieve durable TFR in a larger fraction of CML patients. One recent development has been to invoke the presence of somatic mutations other than BCR::ABL1 to explain primary resistance to TKIs, and disease progression (*Branford et al., 2019*). A complementary perspective is to consider leukemic cell state and heterogeneity as a fundamental determinant of response to TKI to augment the mutational foundation, as has been recently demonstrated in AML (*Zeng et al., 2022*).

Our single-cell multiomics maps show clear differences in overall cell composition within stem and progenitor compartments in leukemia patients at diagnosis versus nBM. Importantly, treatment failures and optimal responders displayed distinctive enrichment of specific cell clusters. Using our 11-molecularly defined clusters as anchors, we deconvoluted the bulk gene expression profiles from n=59 CML patients to infer constituent cell populations and found that a more profound stemness/primitive signature in the BM was consistently associated with inferior therapy response. Indeed, there is growing recognition of the burden of primitive cells, and the ratio of BCR::ABL1+ vs. BCR::ABL1- within the primitive compartment as potentially clinically relevant features such as hemoglobin, blast count, overall survival, progression-free survival, and therapy response (*Fathy El-Metwaly et al., 2021*; *Krishnan et al., 2023*; *Mustjoki et al., 2013*; *Thielen et al., 2016*). Apart from the differential burden, the primitive cells from optimal and sub-optimal responders could also be qualitatively different e.g., in terms of gene expression profile as suggested recently (*Krishnan et al., 2023*). The relevance of BCR::ABL1+ LSC burden to first and second generation TKI therapy outcome stands in contrast to the leukemic progenitors (Ph+ CD34+CD38+) which did not show such a correlation (*Mustjoki et al., 2013*). Moreover, longstanding observations have revealed that CML patients at diagnosis contain both BCR::ABL1+ LSCs vs. BCR::ABL1- HSC within the Lin-CD34+CD38-/low immunophenotypic compartment, and BCR::ABL1+ LSCs suppress residual normal HSCs (*Chen et al., 2023*; *Coulombel et al., 1983*). Predictably, a lower fraction of BCR::ABL1- normal stem cells at diagnosis and during therapy relates to hematological toxicity with delayed or compromised restoration of Phneg hematopoiesis upon successful response to TKIs (*Janssen et al., 2012*). Here, we for the first time present an effective means to separate CML LSCs from HSCs within the stem cell compartment of patients. Prospective optimal responders to TKI treatment had a higher content of BCR::ABL1-CD26-CD35+ cells at diagnosis, and 3 months following therapy versus the treatment failures. We have recently demonstrated that Lin-CD34+CD38-/lowCD45RA-CD35+ cells are at the top of human hematopoietic hierarchy and display chromatin accessibility and transcriptional enhancers in line with their multilineage long-term reconstitution ability (*Sommarin et al., 2021*).

With the high-throughput and resolution now provided by single-cell methods, BCR::ABL1+ cells can be distinguished from BCR::ABL1- cells residing within the same immunophenotypic compartment as described above, and importantly, heterogeneity within BCR::ABL1+ LSC subpopulations in terms of cell surface markers, molecular signature, and TKI response can be measured. The use of CITE-seq enabled the generation of patient-specific maps providing a panoramic yet exquisitely detailed view of cellular heterogeneity. Intriguingly, one of the emerging principles from the single-cell studies has been that not all LSC subpopulations are equally sensitive to BCR::ABL1 inhibition; while a majority of the LSCs are depleted, at least a fraction survives (*Giustacchini et al., 2017*; *Warfvinge et al., 2017*). This is corroborated by an overall reduction in BCR::ABL1+ Lin-CD34+CD38-/low cells but a striking enrichment of cKIT-CD26+ subpopulation displaying primitive and quiescent molecular program upon commencing TKI therapy (*Warfvinge et al., 2017*). Using CITE-seq, we confirmed the existence of these cells already at diagnosis and showed that BCR::ABL1+ vs. BCR::ABL1- can be efficiently discriminated as CD26+CD35- and CD26-CD35+, respectively within the Lin-CD34+CD38-/low fraction, and thus facilitating improved assessment of LSC burden.

The inspection of single-cell maps raises questions both regarding the origin of the interpatient heterogeneity in LSC burden as well as how these cells differentially manage to survive TKI therapy.

Given the heterogeneity within normal HSCs with respect to lineage bias and clonal output (*Haas et al., 2018*), it is tempting to speculate that stem/progenitor cell acquiring the BCR::ABL1 oncogenic hit perhaps might be different across CML patients ensuring disparity in LSC load and characteristics. An equally plausible alternative is variability in the bone marrow niche that might favor differential abundance of stem/lineage-biased progenitors in patients (*Baryawno et al., 2017*). Once established, at least a subset of the LSCs is likely to be independent of BCR::ABL1 kinase signaling for their survival (*Bhatia et al., 2003*; *Corbin et al., 2011*; *Hamilton et al., 2012*). However, whether their self-sustenance involves switching to non-kinase activity of BCR::ABL1 protein, or achieving total BCR::ABL1 independence, and whether such mechanisms are inherently active in treatment naïve cells or become active only after exposure to TKIs is poorly understood (*Zhao and Deininger, 2020*).

We foresee future investigations to provide several key insights. First, accurate and direct detection of fusion transcripts such as BCR::ABL1 in single cells as part of scRNA-seq remains a major constraint, however, new approaches should allow direct measurement of BCR::ABL1 on a massive scale. Second, CITE-seq, a single-cell multiome approach, like any other high-throughput single-cell method comes with methodological, technical, and biological challenges. Cell sample handling, library preparation, sequencing quality and depth, drop-out of lowly expressed genes, and taking a molecular 'snapshot' of cells in time are all merely some factors that could introduce variability and noise to the data. A simpler, faster, and cheaper FACS panel detecting CD26$^+$ and CD35$^+$ surface markers can possibly work as an applicable clinical tool to quantify leukemic and non-leukemic stem cells in CML patients. Future prospective studies of this FACS panel coupled to clinical trials in large patient cohorts will establish the diagnostic and prognostic value of the relative abundance of these populations. It will be importance to evaluate if CD26$^-$CD35$^+$ cells are critical for restoring normal hematopoiesis once the TKI therapy diminishes the leukemic load, and whether patients with low counts of CD26$^-$CD35$^+$ cells at diagnosis have a relatively higher risk of developing hematologic toxicity such as cytopenia during therapy. The relative levels of CD26$^-$ and CD35$^+$ stem cells for TFR will also be an important area of investigation.

Third, joint longitudinal analysis of leukemic as well as immune compartments is likely to be informative, especially for TFR. The observations that patients with activated immune signature at diagnosis are more likely to optimally respond to TKIs (*Radich et al., 2023*) serve as apt reminder for collective analyses. Fourth, our capacity to generate single-cell datasets, ironically outpaces our ability to extract information. Development of tools that allow biologists and clinicians to analyze large-scale datasets without requiring dedicated bioinformatics infrastructure can overcome the challenge (*Dhapola et al., 2022*). Finally, the clonal relationships and lineage output of candidate BCR::ABL1$^+$ LSC subpopulations remain unknown. By coupling sc-multiomics with barcoding analysis and lineage tracing (*Wagner and Klein, 2020*), it should be possible to evaluate, for instance, whether the BCR::ABL1$^+$CD26$^+$ cells detected in patients in TFR are a sub-clone that persists from diagnosis through therapy, and whether it is responsible for recurrence in TFR.

The notion of LSCs has been invoked for a long time in the genetically defined and molecularly targeted paradigm of CML (*Holyoake and Vetrie, 2017*). Still, these entities remain essentially peripheral to the disease management (*Zhao and Deininger, 2021*). We posit that the heterogeneity of LSCs is a barrier towards their efficient measurement and safe purging. Our high-resolution single-cell multiomics maps suggest how to probe and deconstruct heterogeneity of CML, thereby permitting inference of leukemic vs. non-leukemic cells, estimation of BCR::ABL1$^+$ LSCs, enumeration of their molecular features, and prospective isolation. Understanding how the cellular heterogeneity and plasticity emerges in the absence of extensive genetic variability would inform if the fully leukemogenic residual cells could either be safely eliminated pharmacologically or kept perpetually suppressed by an empowered immune system to avoid recurrence (*Hsieh et al., 2021*; *Zhao and Deininger, 2023*). Alternatively, longitudinal sampling and sc-omics combined with barcoding may also reveal if the stem cell subpopulations and states are interconvertible (*Lenaerts et al., 2010*; *Marjanovic et al., 2013*), and, therefore, appear as potentially inexhaustible pool that can only be slowly eroded by several years of TKI treatment in a subset of patients. A more formidable barrier is the unstated but still presumed equivalence of immunophenotype and function. Rather than describing LSCs solely by surface markers, we propose that treating leukemic cellular clones as the fundamental units of selection and evolution during therapy would have a more meaningful impact in predicting response to existing therapy and switch to another TKI. The next line of advances will require assessing LSCs as

pool of clones defined by their ability to contribute to primary and secondary resistance in patients on therapy, and recurrence in TFR without recourse to animal models.

## Materials and methods

### Patient samples

The bone marrow was obtained from patients enrolled in either BosuPeg clinical trial (NCT03831776), NordCML006 (NCT00852566), BFORE (NCT02130557), or from Skåne University Hospital, Lund after informed consent and processed according to the guidelines approved by regional research ethics committees of sites involved in the trial. The information on age, gender, TKIs, BCR::ABL1$^{IS}$ %, and therapy response as per ELN guidelines are provided as *Supplementary file 1*. MNCs or CD34$^+$ cells were enriched from the BM aspirates using magnetic microbeads and subsequently cryopreserved. The collection, processing, and generation of CITE-seq of bone marrow from age-matched healthy donors used in this study has been described previously (*Sommarin et al., 2021*).

### CITE-seq sample preparation and facs sort

Sample preparation was performed according to *Stoeckius et al., 2017* with minor adaptations. In brief, CML MNC, or CD34 enriched BM samples was thawed, FCR blocked 1/5 (130-046-703, Miltenyi) for 10 min, and washed before staining. Samples were stained with a PeCy5 conjugated lineage cocktail (CD2, CD3, CD14, CD16, CD19, CD235a), CD34 FITC, and a subset of the CITE-seq panel for 30 min on ice. Samples were washed and split into two tubes and stained with either CD38-PeCy7 or CD38-CITE-seq antibody followed by individual hashtags and the rest of the CITE-seq panel for 30 min on ice (*Supplementary file 2* and *Supplementary file 3*). Cells were washed and resuspended in PBS, 2% FBS, 1/100 7AAD (Hyclone, 559925, BD Bioscience). Two populations (Lin$^-$CD34$^+$ and Lin$^-$CD34$^+$CD38$^{-/low}$) were sorted per sample using FACSAriaII/Aria III (BD Bioscience) into 300 µl PBS, 0.04% BSA. Sorted cells were centrifuged at 300 × g for 10 min and volume was adjusted to 45 µl.

### CITE-seq library preparation and sequencing

CITE-seq library preparation and sequencing Cells were processed through Chromium Controller (10 x Genomics) with a total of 20–30.000 cells loaded per lane. Single-cell cDNA, HTO (Hashtags), and ADT (CITE-seq antibodies) libraries was prepared using Chromium Single-Cell 3' V3 as per manufacturer instructions (CG000183 Rev C) as reported previously (*Stoeckius et al., 2017*). In brief, to increase HTO and ADT library yield, HTO and ADT-specific primers were spiked-in during the cDNA amplification step. cDNA and HTO/ADT libraries were isolated using 2 X SPRI beads per the manufacturers protocol. HTO and ADT libraries were subjected to adaptor ligation and sample index in separate PCRs using KAPA Hifi PCR Readymix (10 and 11 PCR cycles, respectively). The cDNA library was subjected to fragmentation, end repair, A-tailing, adaptor ligation, and sample index PCR (12 cycles). cDNA, HTO, and ADT libraries were sequenced together (65% cDNA, 30% ADT, 5% HTO) on Illumina sequencers with the following read length configuration: Read1=28, i7=8, i5=0, Read2=91. The raw data was processed using Cell Ranger 3.0.2 with GRCh38 as a reference genome.

### CITE-seq analysis

Cell-gene matrices produced by Cell Ranger were analyzed using Scarf (v.0.18.12 *Dhapola et al., 2022*). In brief, cells with low or high gene count (<1000 or>9000 genes) and cells with high percentage mitochondrial gene UMIs (>11% or >6% for CML samples and healthy BM reference, respectively) were excluded from the analysis. The matrices were demultiplexed using Otsu thresholding (automatized pixel-based thresholding of HTO count per cell, background buffer = 0.1, override value = 0.5 Otsu thresholding was run twice for CML4, CML9. For nBM the thresholds were adjusted to HTO_1=32, HTO_2=18, HTO_3=17). HTO UMAPs were generated through Scarf (dims = 0) and clustered with Paris clustering (n_clusters = 2). If any RNA Leiden cluster overlapped with HTO paris cluster >70% and conversely, it was considered doublets and excluded from analysis.

Post-demultiplexing, for each sample, 2000 highly variable genes (HVGs) were identified (min_cells = 50) with the 'mark_hvgs' function in Scarf and used for PCA (principal component analysis) (CML8; HVGs = 1000, min_cells = 30). KNN (K-Nearest Neighbors) graphs were calculated based on the top 20 principal components (n_neighbours = 11) and utilized to build UMAP embedding of the cells

(min_dist = 1, spread = 2, n_epochs = 2000). Clustering was performed using Lieden clustering (resolution set to 1 and 0.76 for CML and nBM samples, respectively).

Projection of Lin⁻CD34⁺ and Lin⁻CD34⁺CD38⁻/low CML subpopulations.

## Mapping reference

The nBM is merged CITE-seq data of two Lin⁻CD34⁺ FACS sorted bone marrow samples (**Sommarin et al., 2021**) reanalyzed as described above'. Cluster identity was determined by (a) the top 10 genes in 'run_ marker_search' function in Scarf. In brief, each genes expression value is ranked per cell and a gene's mean rank per cluster is divided by the sum of mean to determine the cluster specificity, (b) CLR (centered-log ratio) normed ADT expression across clusters was used to determine the immunophenotypic identity the cells, and (c) cell cycle scoring using the 'run_cell_cycle_scoring' function in Scarf was used to estimate cell cycle phase.

## Label transfer through reference-based cell projection

CML subpopulations were mapped to Lin⁻CD34⁺ nBM using Scarf (v.0.18.12). In short, cells were projected using 'K-Nearest Neighbours (KNN) mapping through Scarf's 'run_mapping' function (neighbors = 5). Cell cluster identity for CML cells was determined using the 'get_target_classes' function with the threshold set to 0.5 (>50% of the total weight score from the five top-matched reference cells must be cluster-specific to assign identity). Mapping scores (how frequently the reference cells end up as one of the top five neighbours of the projected CML cells) for the individual reference cells was calculated per projected sample using 'get_mapping score.'.To visualize the mapping score and compare the mapping across samples, cells size of reference cells was set proportional to the score in the healthy bone marrow reference UMAP.

Following the mapping of Lin⁻CD34⁺CD38⁻/low CML cells (n=8), the primitive clusters from all patients were merged in Scarf. The 'make_graph' function in Scarf was used to identify the primitive CMLs cells top 500 HVGs (min_cells = 30, gender-specific genes excluded). Lin⁻CD34⁺CD38⁻/low healthy bone marrow cells (n=1) were projected onto the healthy reference as described above and the primitive cluster was added to the primitive CML cell clusters in Scarf. The UMAP was generated using the 500 CML-specific HVGs (k=11, PC = 20, n_epoch = 500) and with PCA fitted to the primitive CML cells only using Scarf's 'make graph' function. Subsequently, cells were clustered using the Leiden clustering method (resolution = 0.1).

BCR::ABL1 status of the primitive clusters was determined using two DEG signatures: (1) Normal HSC vs BCR::ABL1⁺, and (2) BCR::ABL1⁻ vs BCR::ABL1⁺ where Lin⁻CD34⁺CD38⁻/lowBCR::ABL1⁺ cells have been validated through single-cell RT-qPCR (**Giustacchini et al., 2017**). Up- and down-regulated genes were defined as log2 fold change >1 and <0, respectively. The expression of signature genes was normalized in Scarf, Z-scored to provide an average value count of the signature per cell and visualized in the UMAP using Scarf.

To visualize the BCR::ABL1⁺ primitive cells alongside the full CML heterogeneity, Lin⁻CD34⁺ and Lin⁻CD34⁺CD38⁻/low CML CITE-seq data from the same patient sample was merged in Scarf (The sub populations were FACS sorted at the time point and sequenced together). UMAPs of the merged data sets were generated as previously described (2000 HVGs, neighbors = 11, PC = 20). Cell indexes from BCR::ABL1⁺ and BCR::ABL1⁻ primitive CML clusters was matched to the Lin⁻CD34⁺CD38⁻/low cell data.

ADT expression of BCR::ABL1⁺ and BCR::ABL1⁻ cells were retrieved from the Zarr file in Scarf. The data thereafter was CLR (centered-log ratio) normalized and the log2 fold change was calculated between the two groups of cells using mean expression. p-values were calculated with the Mann–Whitney U test and they were corrected for multiple hypotheses testing using the Bonferroni method.

## Analysis of differentially expressed genes

DESeq2 was used to identify differentially expressed genes. Significantly up- or down-regulated was selected based on adjusted p-value <0.01 and log2 fold change >1 or < –1. Subsequently, volcano plots showing log2 fold change and -Log10 p-values were generated. Mitochondrial/ribosomal genes, as well as genes with a total count below 10 was excluded from the analysis.

To compare gene expression between clusters 1–11 from Lin⁻CD34⁺ CML cells, n=9 and their normal counterparts, their corresponding normal bone marrow reference clusters (n=2) were randomly pseudo-bulked into three replicates. Cluster-specific signatures genes were defined as significant

up- or down-regulated genes unique to the cluster comparison, and not shared with not any other clusters. CML signature genes were defined as significantly up- or down- regulated genes present in all 11 clusters. Heatmap was generated from the DEseq2 data output (VST), showing the log2 mean expression of all samples per cluster. BCR::ABL1$^+$ and BCR::ABL1$^-$ clusters >25 cells was used for DEG analysis of Lin$^-$CD34$^+$CD38$^{-/low}$ primitive cells (n=7).

### ADT gating

ADT expression was CLR (*centered-log ratio*) normalized in Scarf. The normalized data was transformed into its antilog, multiplied with a scaling factor of 1000, and exported as FCS files. The FCS files was analyzed using FlowJo software. Gated cells ADT scale values were exported as CSV files and matched to cells ADT expression in CITE-seq data to determine original cell identity.

### Gene expression deconvolution

The gene expression counts from CITE-seq molecularly defined populations served as input for CIBERSORTx (*Newman et al., 2019*) at default settings run on its docker version. Deconvolution of publicly available CD34$^+$ bulk gene expression microarray dataset GSE14671 *McWeeney et al., 2010* was performed after robust multichip average (RMA) normalization (*Irizarry et al., 2003*). Upon estimation of relative abundances of subpopulations from CD34$^+$ cells, the values were normalized to 1, the value for each subpopulation represented its percentage/fraction within the query sample. Individual patients were annotated as non-responders or responders to Imatinib therapy as defined in the original study *McWeeney et al., 2010*; non-responders with >65% Ph + metaphases (the patients did not achieve even a minor cytogenetic response as described by the original authors), responders with 0% with Ph + metaphases (achieved CCyR) after 12 months of Imatinib therapy. Notably, the threshold for defining therapy response was proposed by the original authors, and the patients were labeled as either responder or non-responder (*McWeeney et al., 2010*). We adhered to the patient labels provided by the original authors as described in GSE14671.

### Flow cytometry for phenotyping

Lymphoprep kits were used to separate mononuclear cells (MNCs), and magnetic microbeads were used to enrich CD34$^+$ cells (Miltenyi). Antibodies against lineage-specific markers and those for specific populations mentioned in *Supplementary file 3* were used to stain the cells. Sorting and analysis were done using FACS ARIAII/III or LSR FORTESSA (BD Biosciences); data analysis was done with FlowJo (Tree Star).

### qPCR for BCR::ABL1

RNA from FACS-sorted cells was purified using Single-Cell RNA Purification Kit (Norgen, cat# 51800) followed by cDNA synthesis using a High-Capacity cDNA Reverse Transcription Kit (Applied Biosystems, cat# 4368814). Following pre-amplification of cDNA (SsoAdvanced PreAmp Supermix BioRad, cat# 172–5160) for 12 cycles, qPCR was performed with TaqMan gene expression master mix (Applied Biosystems, cat# 4369016) and TaqMan probes for BCR::ABL1 (Assay ID: Hs03024541_ft), GAS2 (Assay ID: Hs00169477_m1 and GAPDH Assay ID: Hs02758991_ft).

### Time to first division in vitro

Lin$^-$CD34$^+$CD38$^{-/low}$CD45RA$^-$CD26$^+$CD35$^-$ or CD26$^-$CD35$^+$ BM single cells from two CML patients were sorted into individual wells of 96-well U-shaped-bottom TPP plates using the FACS Aria III flow cytometer (Becton Dickinson). Cells were sorted directly into 200 µl Stem Span Serum-Free Expansion Medium (SFEM, StemCell Technologies), supplemented with 100 units/ml penicillin, 0.1 mg/ml streptomycin (Hyclone) and the following cytokines from PeproTech: SCF (100 ng/ml), Flt3L (100 ng/ml), TPO (50 ng/ml), and IL7 (10 ng/ml). Cells were visualized and counted in each well thrice a day using an inverted microscope for >140 hr. Dead cells as well as cells that did not divide during this period were excluded from the analysis. The Curve fit-sigmoid approach in GraphPad Prism was used to analyze the data.

## Acknowledgements

We would like to acknowledge patients, study nurses, and other personnel in the clinical centers for their participation in this project. We also thank Clinical Genomics Lund, SciLifeLab and Center for Translational Genomics (CTG), Lund University, for providing expertise and service with sequencing and analysis, and the Lund Stem Cell Center FACS Facility for expert Flow Cytometry technical support. This work was funded by grants from the Swedish Cancer Society, the Ragnar Söderberg Foundation, the Knut and Alice Wallenberg Foundation, the Swedish Research Council, the Swedish Childhood Cancer fund, and a grant from Incyte Biosciences Nordic AB.

## Additional information

### Competing interests

Parashar Dhapola, Göran Karlsson: Is a board member and has equity in Nygen Analytics AB. Henrik Hjorth-Hansen: Received honoraria from Pfizer, Novartis, BMS, and Incyte. Satu Mustjoki: Received honoraria and research funding from BMS, research funding from Novartis, Janpix, and honoraria from Dren Bio. Johan Richter: Received honoraria and research funding from Novartis and Bristol-Myers Squibb (BMS) and honoraria from Ariad. The other authors declare that no competing interests exist.

### Funding

| Funder | Grant reference number | Author |
| --- | --- | --- |
| Swedish Cancer Foundation | | Göran Karlsson |
| Ragnar Söderbergs stiftelse | | Göran Karlsson |
| Knut och Alice Wallenbergs Stiftelse | | Göran Karlsson |
| Vetenskapsrådet | | Göran Karlsson |
| Barncancerfonden | | Göran Karlsson |
| Incyte Biosciences Nordic AB | | Rebecca Warfvinge |

The funders had no role in study design, data collection and interpretation, or the decision to submit the work for publication.

### Author contributions

Rebecca Warfvinge, Conceptualization, Formal analysis, Investigation, Visualization, Methodology, Writing – original draft, Project administration, Writing – review and editing; Linda Geironson Ulfsson, Validation, Investigation, Visualization; Parashar Dhapola, Software, Formal analysis; Fatemeh Safi, Validation; Mikael Sommarin, Methodology; Shamit Soneji, Software, Formal analysis, Validation, Investigation, Visualization; Henrik Hjorth-Hansen, Satu Mustjoki, Johan Richter, Resources; Ram Krishna Thakur, Conceptualization, Supervision, Writing – original draft, Writing – review and editing; Göran Karlsson, Conceptualization, Resources, Supervision, Funding acquisition, Validation, Investigation, Visualization, Methodology, Writing – original draft, Project administration, Writing – review and editing

### Author ORCIDs

Henrik Hjorth-Hansen ⓘ https://orcid.org/0000-0002-2670-5696
Satu Mustjoki ⓘ https://orcid.org/0000-0002-0816-8241
Göran Karlsson ⓘ https://orcid.org/0000-0001-8197-754X

### Ethics

Clinical trial registration NCT03831776, NCT00852566, NCT02130557.
The bone marrow was obtained from patients enrolled in either BosuPeg clinical trial (NCT03831776), NordCML006 (NCT00852566), BFORE (NCT02130557), or from Sk;ne University Hospital, Lund after

informed consent and processed according to the guidelines approved by regional research ethics committees of sites involved in the trial.

Reviewer #1 (Public Review): https://doi.org/10.7554/eLife.92074.3.sa1
Reviewer #3 (Public Review): https://doi.org/10.7554/eLife.92074.3.sa2
Author response https://doi.org/10.7554/eLife.92074.3.sa3

---

## Additional files

### Supplementary files

• Supplementary file 1. Chronic myeloid leukemia (CML) sample cohort.

• Supplementary file 2. Cellular indexing of transcriptomes and epitopes by sequencing (CITE-seq) antibodies.

• Supplementary file 3. FACS antibodies.

• Supplementary file 4. List of marker genes for narrow bone marrow (nBM) clusters.

• Supplementary file 5. List of DEG uniquely changed per cluster in chronic myeloid leukemia (CML) vs. narrow bone marrow (nBM) clusters 1–11 along with their fold change.

• Supplementary file 6. List of DEG along with their fold change between BCR::ABL1$^+$ LSCs vs. BCR::ABL1$^-$ stem cells.

• Supplementary file 7. List of DEG up- and down-regulated in all clusters in chronic myeloid leukemia (CML) vs. narrow bone marrow (nBM) clusters 1–11.

• MDAR checklist

### Data availability

The data from sequencing have been deposited in GEO under accession ID: GSE236233. Previously published data used within this study is available at: GSE173076 (nBM), and GSE14671 (deconvolution). Data and code are available under the following OSF repository: https://osf.io/ns2j8/. The CITE-seq data has been published in an analyzed, ineractive format under the following CellHub repository: https://scarfweb.nygen.io/cellhub/2b2fec2ceb52.

The following datasets were generated:

| Author(s) | Year | Dataset title | Dataset URL | Database and Identifier |
|---|---|---|---|---|
| Warfvinge R, Soneji S, Karlsson G | 2024 | Single cell multi-omics analysis of chronic myeloid leukemia links cellular heterogeneity to therapy response | https://www.ncbi.nlm.nih.gov/geo/query/acc.cgi?acc=GSE236233 | NCBI Gene Expression Omnibus, GSE236233 |
| Dhapola P, Warfvinge R | 2024 | Warfvinge_et_al_2024 | https://osf.io/ns2j8/ | Open Science Framework, ns2j8 |

The following previously published datasets were used:

| Author(s) | Year | Dataset title | Dataset URL | Database and Identifier |
|---|---|---|---|---|
| Sommarin MN, Dhapola P, Karlsson G | 2023 | Single-Cell Multiomics Reveals Distinct Cell States at the Top of the Human Hematopoietic Hierarchy [CITE-seq] | https://www.ncbi.nlm.nih.gov/geo/query/acc.cgi?acc=GSE173076 | NCBI Gene Expression Omnibus, GSE173076 |
| Deininger MW, McWeeney SK | 2010 | Expression signature to predict major cytogenetic response in chronic phase CML patients treated with imatinib | https://www.ncbi.nlm.nih.gov/geo/query/acc.cgi?acc=GSE14671 | NCBI Gene Expression Omnibus, GSE14671 |

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
