## [Editor Report · eLife assessment]

This study presents **fundamental** insights into the heterogeneity of chronic myeloid leukemia (CML) stem cells and their response to tyrosine kinase inhibitor therapy, shedding light on potential mechanisms underlying treatment failure. The study's robust methodology, supported by validation with bulk RNA-seq data and surface marker analysis, provides **compelling** evidence for the identified associations between cellular composition and treatment outcome. These findings contribute to our understanding of CML pathogenesis and may inform the development of more targeted therapeutic strategies.

---

## [Referee Report · Reviewer #1 (Public Review)]

Summary:

This manuscript by Warfvinge et al. reports the results of CITE-seq to generate single-cell multi-omics maps from BM CD34+ and CD34+CD38- cells from nine CML patients at diagnosis. Patients were retrospectively stratified by molecular response after 12 months of TKI therapy using European Leukemia Net (ELN) recommendations. They demonstrate heterogeneity of stem and progenitor cell composition at diagnosis, and show that compared to optimal responders, patients with treatment failure after 12 months of therapy demonstrate increased frequency of molecularly defined primitive cells at diagnosis. These results were validated by deconvolution of an independent previously published dataset of bulk transcriptomes from 59 CML patients. They further applied a BCR-ABL-associated gene signature to classify primitive Lin-CD34+CD38- stem cells as BCR:ABL+ and BCR:ABL-. They identified variability in the ratio of leukemic to non-leukemic primitive cells between patients, showed differences in expression of cell surface markers and determined that a combination of CD26 and CD35 cell surface markers could be used to prospectively isolate the two populations. The relative proportion of CD26-CD35+ (BCR:ABL-) primitive stem cells was higher in optimal responders compared to treatment failures, both at diagnosis and following 3 months of TKI therapy.

Strengths:

The studies are carefully conducted and the results are very clearly presented. The data generated will be a valuable resource for further studies. The strengths of this study are the application of single-cell multi-omics using CITE-Seq to study individual variations in stem and progenitor clusters at diagnosis that are associated with good versus poor outcomes in response to TKI treatment. These results were confirmed by deconvolution of a historical bulk RNAseq data set. Moreover, they are also consistent with a recent report from Krishnan et al. and are a useful confirmation of those results. The major new contribution of this study is the use of gene expression profiles to distinguish BCR-ABL+ and BCR-ABL- populations within CML primitive stem cell clusters and then applying antibody-derived tag (ADT) data to define molecularly identified BCR:ABL+ and BCR-ABL- primitive cells by expression of surface markers. This approach allowed them to show an association between the ratio of BCR-ABL+ vs BCR-ABL- primitive cells and TKI response and study dynamic changes in these populations following short-term TKI treatment.

Weaknesses:

The number of samples studied by CITE-Seq is limited. However, the authors have confirmed their key observations in additional samples. The BCR-ABL+ versus BCR-ABL- status of cells was not confirmed by direct sequencing for BCR-ABL. However, we recognize that the methodologies to perform these analyses on single cells is still evolving and the authors have shown that CD26 and CD35 expression can consistently identify BCR-ABL+ versus BCR-ABL- cells. It will be of interest to learn whether the GEP and surface markers identified here can distinguish BCR-ABL+ primitive stem cells later in the course of TKI treatment.

---

## [Referee Report · Reviewer #3 (Public Review)]

Summary:

In this study, Warfvinge and colleagues use CITE-seq to interrogate how CML stem cells change between diagnosis and after one year of TKI therapy. This provides important insight into why some CML patients are "optimal responders" to TKI therapy while others experience treatment failure. CITE-seq in CML patients revealed several important findings. First, substantial cellular heterogeneity was observed at diagnosis, suggesting that this is a hallmark of CML. Further, patients who experienced treatment failure demonstrated increased numbers of primitive cells at diagnosis compared to optimal responders. This finding was validated in a bulk gene expression dataset from 59 CML patients, in which it was shown that the proportion of primitive cells versus lineage-primed cells correlates to treatment outcome. Even more importantly, because CITE-seq quantifies cell surface protein in addition to gene expression data, the authors were able to identify the BCR/ABL+ and BCR/ABL- CML stem cells express distinct cell surface markers (CD26+/CD35- and CD26-/CD35+, respectively). In optimal responders, BCR/ABL- CD26-/CD35+ CML stem cells were predominant, while the opposite was true in patients with treatment failure. Together, these findings represent a critical step forward for the CML field and may allow more informed development of CML therapies, as well as the ability to predict patient outcomes prior to treatment.

Strengths:

This is an important, beautifully written, well-referenced study that represents a fundamental advance in the CML field. The data are clean and compelling, demonstrating convincingly that optimal responders and patients with treatment failure display significant differences in the proportion of primitive cells at diagnosis, and the ratio of BCR-ABL+ versus negative LSCs. The finding that BCR/ABL+ versus negative LSCs display distinct surface markers is also key and will allow for more detailed interrogation of these cell populations at a molecular level.

Weaknesses:

CITE-seq was performed in only 9 CML patient samples and 2 healthy donors. Additional samples would greatly strengthen the very interesting and notable findings.

---

## [Author Response]

The following is the authors’ response to the original reviews.

**eLife assessment**
This study, utilizing CITE-Seq to explore CML, is considered a useful contribution to our understanding of treatment response. However, the reviewers express concern about the incomplete evidence due to the small sample size and recommend addressing these limitations. Strengthening the study with additional patient samples and validation measures would enhance its significance.

We thank the editors for the assessment of our manuscript. In view of the comments of the three reviewers, we have increased the number of CML patient samples analyzed to confirm all the major findings included in the manuscript. In total, more than 80 patient samples across different approaches have now been analyzed and incorporated in the revised manuscript.

To the best of our knowledge, this is the first single cell multiomics report in CML and differs substantially from the recent single cell omics-based reports where single modalities were measured one at a time (Krishnan et al., 2023; Patel et al., 2022). Thus, the sc-multiomic investigation of LSCs and HSCs from the same patient addresses a major gap in the field towards managing efficacy and toxicity of TKI treatment by enumerating CD26+CD35- LSCs and CD26-CD35+ HSCs burden and their ratio at diagnosis vs. 3 months of therapy. The findings suggest design of a simpler and cheaper FACS assay to simultaneously stratify CML patients for TKI efficacy as well as hematologic toxicity.

**Reviewer 1:**
Summary:This manuscript by Warfvinge et al. reports the results of CITE-seq to generate singlecell multi-omics maps from BM CD34+ and CD34+CD38- cells from nine CML patients at diagnosis. Patients were retrospectively stratified by molecular response after 12 months of TKI therapy using European Leukemia Net (ELN) recommendations. They demonstrate heterogeneity of stem and progenitor cell composition at diagnosis, and show that compared to optimal responders, patients with treatment failure after 12 months of therapy demonstrate increased frequency of molecularly defined primitive cells at diagnosis. These results were validated by deconvolution of an independent previously published dataset of bulk transcriptomes from 59 CML patients. They further applied a BCR-ABL-associated gene signature to classify primitive Lin-CD34+CD38- stem cells as BCR:ABL+ and BCR:ABL-. They identified variability in the ratio of leukemic to non-leukemic primitive cells between patients, showed differences in the expression of cell surface markers, and determined that a combination of CD26 and CD35 cell surface markers could be used to prospectively isolate the two populations. The relative proportion of CD26-CD35+ (BCR:ABL-) primitive stem cells was higher in optimal responders compared to treatment failures, both at diagnosis and following 3 months of TKI therapy.Strengths:The studies are carefully conducted and the results are very clearly presented. The data generated will be a valuable resource for further studies. The strengths of this study are the application of single-cell multi-omics using CITE-Seq to study individual variations in stem and progenitor clusters at diagnosis that are associated with good versus poor outcomes in response to TKI treatment. These results were confirmed by deconvolution of a historical bulk RNAseq data set. Moreover, they are also consistent with a recent report from Krishnan et al. and are a useful confirmation of those results. The major new contribution of this study is the use of gene expression profiles to distinguish BCRABL+ and BCR-ABL- populations within CML primitive stem cell clusters and then applying antibody-derived tag (ADT) data to define molecularly identified BCR:ABL+ and BCR-ABL- primitive cells by expression of surface markers. This approach allowed them to show an association between the ratio of BCR-ABL+ vs BCR-ABL- primitive cells and TKI response and study dynamic changes in these populations following short-term TKI treatment.Weaknesses:One of the limitations of the study is the small number of samples employed, which is insufficient to make associations with outcomes with confidence. Although the authors discuss the potential heterogeneity of primitive stem, they do not directly address the heterogeneity of hematopoietic potential or response to TKI treatment in the results presented. Another limitation is that the BCR-ABL + versus BCR-ABL- status of cells was not confirmed by direct sequencing for BCR-ABL. The BCR-ABL status of cells sorted based on CD26 and CD35 was evaluated in only two samples. We also note that the surface markers identified were previously reported by the same authors using different single-cell approaches, which limits the novelty of the findings. It will be important to determine whether the GEP and surface markers identified here are able to distinguish BCR-ABL+ and BCR-ABL- primitive stem cells later in the course of TKI treatment. Finally, although the authors do describe differential gene expression between CML and normal, BCR:ABL+ and BCR:ABL-, primitive stem cells they have not as yet taken the opportunity to use these findings to address questions regarding biological mechanisms related to CML LSC that impact on TKI response and outcomes.
**Reviewer #1 (Recommendations For The Authors):**
Minor comment: Fig 4 legend -E and F should be C and D.

We thank the reviewer for positive assessment of our work. Here, we highlight the updates in the revised manuscript considering the feedback received.

Minor comment: Fig 4 legend -E and F should be C and D.

We have edited the revised manuscript accordingly

One of the limitations of the study is the small number of samples employed, which is insufficient to make associations with outcomes with confidence.

Although we performed CITE-seq for 9 CML patient samples at diagnosis, we extended our investigations to include additional samples (e.g., largescale deconvolution analysis of samples, Fig 3 C-E, qPCR for BCR::ABL1 status, Fig. 6A, and the ratio between CD35+ and CD26+ populations at diagnosis and during TKI therapy, Fig. 6C-D) as described in the manuscript.

In comparison to a scRNA-seq, multiomic CITE-seq involves preparation and sequencing of separate libraries corresponding to RNA and ADTs thereby being even more resource demanding limiting our capacity to process an extensive number of patient samples. To confirm our findings in a larger cohort we have therefore adopted a computational deconvolution approach, CIBERSORT to analyze a larger number of independent samples (n=59). This reflects a growing, sustainable trend to study larger number of patients in face of still prohibitively expensive but potentially insightful scomics approaches (For example, please see Zeng et al, A cellular hierarchy framework for understanding heterogeneity and predicting drug response in acute myeloid leukemia, Nature Medicine, 2022).

However, in view of the comment, we have now substantially increased the number of analyzed patients in the revised manuscript. These include increased number of patient samples to investigate the ratio between CD35 and CD26 marked populations at diagnosis, and 3 months of TKI therapy (from n=8 to n=12 with now 6 optimal responders and 5 treatment failure at diagnosis and after TKI therapy), qPCR for BCR::ABL1 expression status at diagnosis (from n=3 to n=9) , and followed up the BCR::ABL1 expression in three additional samples after TKI therapy. Moreover, we examined the CD26 and CD35 marked populations for expression of GAS2, one of our top candidate LSC signature genes in three additional samples at diagnosis and at 3m follow up. Thus, >80 patient samples across different approaches have been analyzed to strengthen all major conclusions of the study.

We emphasize that we were cautious in generalizing the observation obtained from any one approach and sought to confirm any major finding using at least one complementary method. As an example, although CITE-seq (n=9) showed altered frequency of all cell clusters between optimal and poor responders (Fig. 3B), we refrained from generalizing because our independent large-scale computational deconvolution analysis (n=59) only substantiated the altered proportion of primitive and myeloid cell clusters (Fig. 3E).

Although the authors discuss the potential heterogeneity of primitive stem, they do not directly address the heterogeneity of hematopoietic potential or response to TKI treatment in the results presented.

Thanks for noting the discussion on heterogeneity of the primitive stem cells. As described in the original manuscript, the figure 6 D-E showed a relationship between heterogeneity and TKI therapy response. The results showed that CD35+/CD26+ ratio within the HSC fraction associated with this therapy response. We have now increased the number of patient samples analyzed and present the updated results in the revised manuscript (now figure 6 C-D). These observations set the stage for assessing whether long term therapy outcome can also be influenced by heterogeneity at diagnosis.

We have shown the hematopoietic potential of HSCs marked by CD35 expression in an independent parallel study and therefore only mentioned it concisely in the current manuscript. A combination of scRNA-seq, scATAC-seq and cell surface proteomics showed CD35+ cells at the apex of healthy human hematopoiesis, containing an HSCspecific epigenetic signature and molecular program, as well as possessing self-renewal capacity and multilineage reconstitution in vivo and vitro. The preprint is available as Sommarin et al. ‘Single-cell multiomics reveals distinct cell states at the top of the human hematopoietic hierarchy’, Biorxiv;https://www.biorxiv.org/content/10.1101/2021.04.01.437998v2.full

We also note that the surface markers identified were previously reported by the same authors using different single-cell approaches, which limits the novelty of the findings.

Our current manuscript is indeed a continuation of and builds onto our previous paper (Warfvinge R et al. Blood, 2017). In contrast to our previous report which was limited to examination of only 96 genes per cell, CITE-seq allowed us to examine the molecular program of cells using unbiased global gene expression profiling. Finally, although CD26 appears, once again as a reliable marker of BCR::ABL1+ primitive cells, CD35 emerges as a novel and previously undescribed marker of BCR::ABL1- residual stem cells. A combination of CD35 and CD26 allowed us to efficiently distinguish between the two populations housed within the Lin-34+38/low stem cell immunophenotype.

Another limitation is that the BCR-ABL + versus BCR-ABL- status of cells was not confirmed by direct sequencing for BCR-ABL. The BCR-ABL status of cells sorted based on CD26 and CD35 was evaluated in only two samples

Single cell detection of fusion transcripts is challenging with low detection sensitivity in single cell RNA-seq as has been noted previously (Krishnan et al. Blood, 2023, Giustacchini et al. Nature Medicine, 2017, Rodriguez-Meira et al. Molecular Cell, 2019). However, this is likely to change with the inclusion of targetspecific probes in scRNA-seq library preparation protocols. Nonetheless, in view of the comment, we have included more patient samples from the previous n=3 to current n=10 (including TKI treated samples) for direct assessment of BCR-ABL1 status by qPCR analysis; the updated results are included in the revised manuscript (Figure 6A).

It will be important to determine whether the GEP and surface markers identified here are able to distinguish BCR-ABL+ and BCR-ABL- primitive stem cells later in the course of TKI treatment.

We performed qPCR to check for BCR::ABL1 status, and the level of GAS2, one of the top genes expressed in CML cells within CD26+ and CD35+ cells at diagnosis and following 3 months of TKI therapy. The results showed that while CD26+ are BCR::ABL1+, the CD35+ cells are BCR::ABL1- at both time points.Moreover, the expression of LSC-specific gene, GAS2 was specific to BCR::ABL1+ CD26+ cells at both diagnosis as well as following 3 months of TKI therapy. The new results are presented in figure 6B in the revised manuscript.

Finally, although the authors do describe differential gene expression between CML and normal, BCR:ABL+ and BCR:ABL-, primitive stem cells they have not as yet taken the opportunity to use these findings to address questions regarding biological mechanisms related to CML LSC that impact on TKI response and outcomes.

We agree with the reviewer that our major focus here was to characterize the cellular heterogeneity coupled to treatment outcome and therefore we did not delve deep into the molecular mechanisms underlying TKI response. However, in response to this comment, as mentioned above, we noted that one of the top genes in BCR::ABL1 cells (Fig. 4 C; right; in red), GAS2 (Growth Specific Arrest 2) was expressed at both diagnosis and TKI therapy within CD26+ cells relative to CD35+ cells (updated figure 6B). Interestingly, GAS2 was also detected in CML LSCs in a recent scRNA-seq study (Krishnan et al. Blood, 2023) suggesting GAS2 upregulation could be a consistent molecular feature of CML cells. GAS2 has been previously noted as deregulated in CML (Janssen JJ et al. Leukemia, 2005, Radich J et al, PNAS, 2006), control of cell cycle, apoptosis, and response to Imatinib (Zhou et al. PLoS One, 2014). Future investigations are warranted to assess whether GAS2 could play a role in the outcome of long-term TKI therapy.

**Reviewer 2:**
Summary:The authors use single-cell "multi-comics" to study clonal heterogeneity in chronic myeloid leukemia (CML) and its impact on treatment response and resistance. Their main results suggest (1) Cell compartments and gene expression signatures both shared in CML cells (versus normal), yet (2) some heterogeneity of multiomic mapping correlated with ELN treatment response; (3) further definition of s unique combination of CD26 and CD35 surface markers associated with gene expression defined BCR::ABL1+ LSCs and BCR::ABL1- HSCs. The manuscript is well-written, and the method and figures are clear and informative. The results fit the expanding view of cancer and its therapy as a complex Darwinian exercise of clonal heterogeneity and the selective pressures of treatments.Strengths:Cutting-edge technology by one of the expert groups of single-cell 'comics.Weaknesses:Very small sample sizes, without a validation set. The obvious main problem with the study is that an enormous amount of results and conjecture arise from a very small data set: only nine cases for the treatment response section (three in each of the ELN categories), only two normal marrows, and only two patient cases for the division kinetic studies. Thus, it is very difficult to know the "noise" in the system - the stability of clusters and gene expression and the normal variation one might expect, versus patterns that may be reproducibly study artifact, effects of gene expression from freezing-thawing, time on the bench, antibody labeling, etc. This is not so much a criticism as a statement of reality: these elegant experiments are difficult, timeconsuming, and very expensive. Thus in the Discussion, it would be helpful for the authors to just frankly lay out these limitations for the reader to consider. Also in the Discussion, it would be interesting for the authors to consider what's next: what type of validation would be needed to make these studies translatable to the clinic? Is there a clever way to use these data to design a faster/cheaper assay?

We thank the reviewer for appraisal of our manuscript. We take the opportunity to point out the updates in the revised manuscript in view of the comments.

Very small sample sizes, without a validation set. The obvious main problem with the study is that an enormous amount of results and conjecture arise from a very small data set: only nine cases for the treatment response section (three in each of the ELN categories), only two normal marrows, and only two patient cases for the division kinetic studies.

As the reviewer has noted the single cell omics experiments remain resource demanding thereby placing a limitation on the number of patients analyzed. As described above in response to the comments from reviewer 1, multiomic CITE-seq allows extraction of two modalities in comparison to a typical scRNA-seq, however, this also makes it even more limited in the number of samples processed in a sustainable way. This was one of the motivations to analyze a larger number of independent samples (n=59) while benefiting from the insights gained from CITE-seq (n=9). Furthermore, by analyzing CD34+ cells from bone marrow and peripheral blood of CML patients, including both responders and non-responders after one year of Imatinib therapy, we were able to significantly diversity the patient pool, which was lacking in our CITE-seq patient pool. As mentioned above, this reflects a growing trend to analyze larger number of patients while anchoring the analysis on prohibitively expensive but potentially insightful sc-omics approaches (For example, please see Zeng et al, A cellular hierarchy framework for understanding heterogeneity and predicting drug response in acute myeloid leukemia, Nature Medicine, 2022).

As emphasized above, we frequently sought to confirm the findings from one approach using a complementary method and independent samples. For example, although CITE-seq (n=9) showed altered frequency of all cell clusters between optimal and poor responders (Fig. 3B), we refrained from generalizing because an independent largescale computational deconvolution analysis (n=59) only substantiated the altered proportion of primitive and myeloid clusters.

In view of the comment, we have now increased the number of patients analyzed during the revision process. These include increased numbers to investigate the ratio between CD35+ and CD26+ populations at diagnosis, as well as 3 months of TKI therapy, qPCR for BCR::ABL1, and patients examined for GAS2, one of the top genes expressed in CML cells (see response to reviewer 1 for details). Altogether, >80 patient samples across different approaches were analyzed to strengthen the conclusions.

During the revision, we have analyzed cells from 8 CML patients for cell cycle using gene activity scores. This is in addition to the cell division kinetics data reported previously are now together described in the supplementary figures 9C-F.

It is very difficult to know the "noise" in the system - the stability of clusters and gene expression and the normal variation one might expect, versus patterns that may be reproducibly study artifact, effects of gene expression from freezing-thawing, time on the bench, antibody labeling, etc. This is not so much a criticism as a statement of reality: these elegant experiments are difficult, time-consuming, and very expensive.Thus in the Discussion, it would be helpful for the authors to just frankly lay out these limitations for the reader to consider.

We agree with the reviewer that sc-omics approaches can be noisy despite continuing efforts to denoise single cell datasets through both experimental and bioinformatic innovations. Therefore, we have updated the discussion as recommended by the reviewer (paragraph 5 in the discussion).

We also note that CITE-seq, in contrast to scRNA-seq alone provides dual features: surface marker/protein as well as RNA for annotating the same cluster. In our manuscript, for example, cell clusters in UMAP for normal BM; Fig 1B were described using both surface markers (Fig. 1C) and RNA (Fig. 1D) making the cluster identity robust. To further elaborate this approach, a new supplementary figure 1C shows annotations of clusters using both RNA and surface markers.

To potentially address the issue of stability of clusters and gene expression, we compared the marker genes for major clusters from nBM from this study (supplementary table 4, Warfvinge et al.) with those described recently in a scRNA-seq study by Krishnan et al. supplementary table 8, Blood, 2023 using Cell Radar, a tool that identifies and visualizes which hematopoietic cell types are enriched within a given gene set (description: https://github.com/KarlssonG/cellradar; Direct link: https://karlssong.github.io/cellradar/). To compare, we used our in-house gene list for the major clusters as well as mapped the same number of top marker genes based on log2FC from corresponding cluster from Krishnan et al. as inputs to Cell Radar. The Cell Radar plot outputs are shown below.

This approach showed broad similarities across clusters from this study with their counterparts from the other study suggesting the cluster identities reported here are likely to be robust. Please note these figures are for reviewer response only and not included in the final manuscript.

Also in the Discussion, it would be interesting for the authors to consider what's next: what type of validation would be needed to make these studies translatable to the clinic? Is there a clever way to use these data to design a faster/cheaper assay?

Our findings on CD26+ and CD35+ surface markers to enrich BCR::ABL1+ and BCR::ABL1- cells suggest a simpler, faster and cheaper FACS panel can possibly quantify leukemic and non-leukemic stem cells in CML patients. We anticipate that future investigations, clinical studies might examine whether CD26CD35+ cells could be plausible candidates for restoring normal hematopoiesis once the TKI therapy diminishes the leukemic load, and whether patients with low counts of CD35+ cells at diagnosis have a relatively higher chance of developing hematologic toxicity such as cytopenia during therapy.

We briefly mentioned this possibility in the discussion; however, we have now moved it to another paragraph to highlight the same. Please see paragraph 5 in the revised manuscript.

**Reviewer 3:**
Summary:In this study, Warfvinge and colleagues use CITE-seq to interrogate how CML stem cells change between diagnosis and after one year of TKI therapy. This provides important insight into why some CML patients are "optimal responders" to TKI therapy while others experience treatment failure. CITE-seq in CML patients revealed several important findings. First, substantial cellular heterogeneity was observed at diagnosis, suggesting that this is a hallmark of CML. Further, patients who experienced treatment failure demonstrated increased numbers of primitive cells at diagnosis compared to optimal responders. This finding was validated in a bulk gene expression dataset from 59 CML patients, in which it was shown that the proportion of primitive cells versus lineage-primed cells correlates to treatment outcome. Even more importantly, because CITE-seq quantifies cell surface protein in addition to gene expression data, the authors were able to identify that BCR/ABL+ and BCR/ABL- CML stem cells express distinct cell surface markers (CD26+/CD35- and CD26-/CD35+, respectively). In optimal responders, BCR/ABL- CD26-/CD35+ CML stem cells were predominant, while the opposite was true in patients with treatment failure. Together, these findings represent a critical step forward for the CML field and may allow more informed development of CML therapies, as well as the ability to predict patient outcomes prior to treatment.Strengths:This is an important, beautifully written, well-referenced study that represents a fundamental advance in the CML field. The data are clean and compelling, demonstrating convincingly that optimal responders and patients with treatment failure display significant differences in the proportion of primitive cells at diagnosis, and the ratio of BCR-ABL+ versus negative LSCs. The finding that BCR/ABL+ versus negative LSCs display distinct surface markers is also key and will allow for a more detailed interrogation of these cell populations at a molecular level.Weaknesses:CITE-seq was performed in only 9 CML patient samples and 2 healthy donors. Additional samples would greatly strengthen the very interesting and notable findings.
**Reviewer #3 (Recommendations For The Authors):**
My only recommendation is to bolster findings with additional CML and healthy donor samples.CITE-seq was performed in only 9 CML patient samples and 2 healthy donors. Additional samples would greatly strengthen the very interesting and notable findings.

We thank the reviewer for the positive assessment of our manuscript. As mentioned in response to comments from reviewer 1 and 2, CITE-seq remains an reource consuming single cell method potentially limiting the number of patients to be analyzed. However, during the revision process, we have increased the number of patient material analyzed for other assays; these include increased number to investigate the ratio between CD35+ and CD26+ populations at diagnosis, and 3 months of TKI therapy, qPCR for BCR::ABL1, and patients examined for GAS2, one of the top genes expressed in CML cells. Thus, >80 patient samples across different assays have been analyzed to strengthen the conclusions. (Please see comment to reviewer 1 for more details)